# Position: Multi-Agent Explainability Needs Contracts Before Methods

Hak Hyun Kim [1]   Benjamin Huh [1]   Soroush Vosoughi [1]

## Abstract

Multi-Agent Systems (MAS) are deployed at unprecedented scale—from warehouse robot fleets to autonomous vehicle networks to collaborative LLM agents—yet methods for explaining their behavior remain fragmented and underspecified. We analyze 2,381 MAS-related papers from top machine learning venues (2021–2025) and find systematic gaps: 65% omit stakeholder specifications, 76% lack quantitative evaluation bounds, and 99% ignore auditability requirements. These gaps render current MAS XAI research non-comparable, non-reproducible, and disconnected from deployment requirements. We argue that MAS XAI research requires explicit specification of two contracts before developing methods. The **Research Contract** defines six elements: explanandum, stakeholder, intervention unit, evaluation bounds, adversarial context, auditability. The **Agent Contract** defines expected behaviors through obligations, permissions, prohibitions, violation criteria, and accountability chains—providing the baseline against which deviations are explained. These contracts are method-agnostic and architecture-agnostic, applicable to LLM-based, learning-based, and hybrid MAS. Through case studies spanning warehouse robotics, autonomous vehicles, and LLM agent systems, we demonstrate how these contracts can transform vague accounts of multi-agent behavior into explanations that are verifiable, actionable, and comparable. We call on researchers to adopt contracts in their work, conferences to encourage specification in submissions, and platforms to integrate contract templates into MAS benchmarks.

[1]Department of Computer Science, Dartmouth College, Hanover, NH, USA. Correspondence to: Soroush Vosoughi <Soroush.Vosoughi@dartmouth.edu>.

*Proceedings of the 43rd International Conference on Machine Learning*, Seoul, South Korea. PMLR 306, 2026. Copyright 2026 by the author(s).

## 1. Introduction

Multi-Agent Systems (MAS) are experiencing unprecedented growth across research and industry. LLM-based multi-agent frameworks such as AutoGPT, MetaGPT, and CrewAI have collectively surpassed 300,000 GitHub stars as of May 28, 2026 (Significant Gravitas, 2023; FoundationAgents, 2023; crewAIInc, 2023). Enterprise adoption surveys report broad uptake: PwC finds 79% of surveyed firms already adopting AI agents (PwC, 2025), while Capgemini reports meaningful deployment and pilot activity among large enterprises (Capgemini Research Institute, 2025). Beyond software agents, physical multi-agent deployments have expanded at remarkable scale: Amazon operates more than 1 million robots across its operations network (Greenawalt, 2025), and Waymo reports more than 250,000 paid trips each week in the U.S. (The Waymo Team, 2025). The Ocado warehouse incident further illustrates the safety stakes of tightly coupled robotic coordination (Bateman, 2021).

As these systems grow in complexity and real-world impact, explaining their behavior becomes critical—both for debugging during development and for accountability after deployment. The EU AI Act, with high-risk transparency obligations applying from 2 August 2026, mandates that high-risk AI systems "shall be designed and developed in such a way as to ensure that their operation is sufficiently transparent to enable deployers to interpret a system's output" (European Parliament and Council of the European Union, 2024). Similar requirements are emerging in the U.S. through FDA draft guidance for AI-enabled device software functions (U.S. Food and Drug Administration, 2025) and NIST's AI Risk Management Framework (National Institute of Standards and Technology, 2023b).

Yet when multi-agent systems fail, explanations prove elusive. In August 2023, ten Cruise robotaxis simultaneously froze in San Francisco's North Beach district, blocking traffic for 15 minutes (Mitchell, 2023). The official explanation—"cellular connectivity loss due to a music festival 4 miles away"—identified the proximate cause but left fundamental questions unanswered: Why did all vehicles fail simultaneously? Which vehicle should have yielded first? What coordination protocol broke down? Similarly, when an Ocado warehouse fire resulted from a three-robot collision in 2021, the company acknowledged that "the col-

lision of three bots on the grid" caused the fire but never disclosed what coordination algorithm failure allowed high-speed robots to collide (Bateman, 2021). In multi-agent LLM systems, a July 2025 Replit incident reported that an AI coding agent ignored explicit "code freeze" instructions, deleted a production database, and then reportedly misrepresented recovery status—behavior that emerged from multi-system interactions rather than any single component failure (Tyson, 2025).

These incidents reveal a fundamental challenge: explaining multi-agent behavior requires frameworks that current research lacks. Unlike single-agent systems where explainability methods attribute outcomes to input features of one decision-maker, MAS exhibit *emergent behaviors* arising from agent interactions that cannot be predicted from individual agent analysis (Altmann et al., 2024), *distributed causation* where outcomes result from interacting agents' joint decisions rather than any single agent (Gyevnar et al., 2024), and *credit assignment complexity* that grows combinatorially with agent count and interaction horizon (Li et al., 2021; Foerster et al., 2018).

We argue that these challenges stem not from inadequate XAI methods but from a more fundamental problem: **MAS XAI research currently operates without explicit specifications for what should be explained, to whom, and against what baseline of expected behavior**. Current work is non-comparable (due to different implicit assumptions), non-reproducible (no shared evaluation criteria), and disconnected from deployment requirements (governance needs are systematically ignored). The detailed findings in Section 4 show that most papers lack specifications for target stakeholders, evaluation bounds, and accountability—the very elements needed for explanations to be validated, compared, and trusted.

**Position.** We argue that MAS XAI research requires explicit specification of two contracts before proposing explanation methods:

1. A **Research Contract** defining the explanandum (what to explain), stakeholder (for whom), counterfactual intervention unit (at what granularity), evaluation criteria with quantitative bounds, adversarial context, and governance requirements.

2. An **Agent Contract** defining expected agent behaviors through obligations, permissions, prohibitions, violation criteria, and accountability chains—providing the behavioral baseline against which deviations can be identified and attributed.

These contracts are method-agnostic: they specify *what* must be achieved, not *how* to achieve it. Any XAI technique—whether attention-based, counterfactual, or decomposition-based—can satisfy a well-specified contract, but without such specification, even technically sophisticated methods produce explanations that cannot be validated, compared, or trusted. Because specification norms propagate through the venues that set community standards, we deliberately ground this proposal at a top machine learning conference: if contracts are to become routine practice, adoption must begin where review processes and publication norms shape what the field considers rigorous.

This paper makes four contributions: (1) we propose the **Research Contract** framework with six essential elements that researchers should specify before developing XAI methods (§3.1); (2) we propose the **Agent Contract** framework that is architecture-agnostic across LLM-based, learning-based, and hybrid MAS, providing behavioral baselines for explanation (§3.2); (3) we empirically validate the prevalence of specification gaps through systematic analysis of 2,381 MAS papers, demonstrating that current research lacks the specifications our framework requires (§4); and (4) we demonstrate through illustrative case studies how these contracts can transform vague post-hoc rationalizations into verifiable, actionable, and comparable explanations (Appendix A).

## 2. Background and Related Work

### 2.1. Explainable AI for Multi-Agent Systems

Explainable AI (XAI) has matured significantly for single-agent systems, with established taxonomies distinguishing intrinsic interpretability from post-hoc explanation, and local from global explanations (Molnar, 2020). However, extending these methods to multi-agent systems introduces fundamental challenges that surveys have only recently begun to address systematically.

Qing et al. (2022) provides a comprehensive XRL survey, proposing an RL-centric taxonomy with four explanation targets: agent model, reward, state, and task. Their multi-agent coverage includes MAVIPER for policy extraction and Shapley-based reward decomposition, but acknowledges that "interpreting the decisions of multiple agents over time is combinatorially more complex than understanding individual, static decisions." Recent MAS safety analyses similarly emphasize that local specification quality does not automatically compose to safe system-level behavior, because interaction dynamics introduce additional opacity and failure modes (Altmann et al., 2024).

Recent work has begun addressing specific MAS explanation challenges. Gyevnar et al. (2024) demonstrate that causal counterfactual simulation can recover interaction-level causes behind agent decisions in sequential MAS. Rodriguez et al. (2025) propose requirements-based explainability, tracing agent behavior back to original specifications

through the TriQPAN design pattern—a shift from post-hoc explanation to design-time traceability. Alzetta et al. (2020) introduce Real-Time BDI (RT-BDI) for time-critical MAS, emphasizing that explanations must meet temporal deadlines in safety-critical applications.

Despite these advances, the field lacks a unifying framework. Existing surveys either focus on single-agent XRL with peripheral multi-agent coverage (Qing et al., 2022; Milani et al., 2024), or address specific challenges (causal interaction effects, real-time constraints) without integration. No survey or position paper has proposed *what specifications* MAS XAI research should provide before developing methods.

## 2.2. Normative Multi-Agent Systems and Contracts

The literature on normative MAS provides a 25-year foundation for specifying agent behavior through obligations, permissions, and prohibitions. Electronic institutions (Esteva et al., 2001) formalize institutional governance through roles, norms, and speech acts, with AMELI middleware (Esteva et al., 2005) providing runtime enforcement. Social commitment theory (Singh, 1999; Castelfranchi, 1995) models directed obligations between agents: $C(\text{debtor}, \text{creditor}, \phi)$ specifies that the debtor commits to the creditor to make $\phi$ true. The MOISE+ organizational model (Hübner et al., 2007) integrates structural, functional, and deontic specifications, enabling role-based behavior constraints with explicit sanctions.

More recently, Boella & van der Torre (2006) extended Design by Contract (Meyer, 1992) to multi-agent systems through deontic logic, expressing preconditions and postconditions as directed obligations: $O_{a,b}(\phi)$ denotes agent $a$'s obligation to $b$ to achieve $\phi$. Morales et al. (2013) demonstrated automated norm synthesis through the IRON system, which monitors MAS execution, detects coordination conflicts, and synthesizes candidate norms online. Contemporary governance standards and regulations emphasize traceability and accountability chains that are auditable across actors and the system lifecycle (European Parliament and Council of the European Union, 2024; International Organization for Standardization, 2023).

This normative tradition provides formal machinery for specifying *what agents should do*, but has remained largely separate from the XAI literature, which focuses on explaining *what agents actually did*. Our Agent Contract proposal bridges this gap: normative specifications define expected behavior, against which deviations can be identified and explained.

## 2.3. Documentation Standards as Precedent

The AI/ML community has successfully established documentation standards that transformed research practice. Model Cards (Mitchell et al., 2019) provide structured templates documenting model performance across demographic groups, intended uses, and limitations; they are now integrated into platform documentation workflows such as the Hugging Face Hub (Hugging Face, 2026). Datasheets for Datasets (Gebru et al., 2021) systematize data documentation through lifecycle-phase questions covering motivation, composition, collection, and maintenance. Data Statements (Bender & Friedman, 2018) extend this to NLP with domain-specific elements (speaker demographics, annotator characteristics, language variety). IBM AI FactSheets (IBM, 2018) provide enterprise-grade automation, capturing metadata during training for governance dashboards.

These frameworks succeeded through common mechanisms: they addressed acute problems researchers faced such as model opacity and data provenance, they provided simple templates rather than heavy formalism, they integrated with existing infrastructure like Hugging Face and training frameworks, and they enabled community-driven evolution rather than top-down mandates. Regulatory bodies subsequently integrated these documentation practices: the NIST AI RMF Playbook explicitly references Model Cards, Datasheets for Datasets, and Data Statements (National Institute of Standards and Technology, 2023a), while ISO/IEC 42001 reinforces structured documentation and governance processes (International Organization for Standardization, 2023).

The success of these documentation standards demonstrates that research communities can adopt standardized specifications when they provide immediate value. Our Research Contract and Agent Contract proposals follow this precedent: structured templates that researchers complete *before* developing XAI methods, enabling comparison, reproduction, and validation.

## 2.4. Missing Specification Standards for MAS XAI

Despite advances in MAS explainability methods, normative specifications, and documentation standards, **to our knowledge, no existing work proposes what MAS XAI research should specify before developing explanation methods**. Current MAS XAI papers typically develop explanation techniques without specifying target stakeholders or their information needs, evaluate against implicit baselines without defining expected agent behavior, report aggregate metrics without quantitative bounds on explanation quality, and omit adversarial considerations and auditability requirements.

This gap affects not only deployment but also academic

*Table 1.* Research Contract elements for MAS XAI. Each element addresses a specific question that current research often leaves implicit.

| Element | Question Addressed | Derivation & Rationale |
|---|---|---|
| **Explanandum** | What is being explained? | XAI foundations: explanations require a defined target (decision, behavior, outcome, coordination pattern) (Miller, 2019) |
| **Stakeholder** | For whom? | Social XAI: different audiences require different explanations; developers ≠ regulators ≠ end-users (Tomsett et al., 2018) |
| **Intervention Unit** | At what granularity? | MAS-specific: explanations can target individual actions, agent policies, pairwise interactions, or system-level emergent behaviors |
| **Evaluation Bounds** | How good is sufficient? | Scientific validity: claims require quantitative criteria; Model Cards require performance metrics (Mitchell et al., 2019) |
| **Adversarial Context** | Under what attacks? | Robustness: explanations can be manipulated; adversarial XAI is active research area (Slack et al., 2020) |
| **Auditability** | What governance? | Regulatory: EU AI Act Article 12 mandates logging; Article 13 mandates transparency (European Parliament and Council of the European Union, 2024) |

rigor. Without shared specifications, researchers cannot compare methods (different papers explain different things to different audiences), reproduce results (evaluation criteria are implicit), or validate that explanations serve their intended purposes (no stakeholder requirements defined). The following sections propose Research Contracts and Agent Contracts to fill this gap.

## 3. The Framework: Research and Agent Contracts

MAS XAI benefits from two complementary contracts: a **Research Contract** specifying what explanations should achieve, and an **Agent Contract** specifying expected behaviors against which deviations are explained.

### 3.1. Research Contract

The Research Contract specifies six elements that researchers should define before developing MAS XAI methods. These elements derive from three sources: (1) XAI literature identifying explanation requirements (Miller, 2019; Molnar, 2020), (2) regulatory frameworks mandating transparency and accountability (European Parliament and Council of the European Union, 2024; National Institute of Standards and Technology, 2023b), and (3) documentation standards demonstrating successful research specification (Mitchell et al., 2019; Gebru et al., 2021). Table 1 summarizes each element.

**Formal Research Contract.** We define the Research Contract as:

$$\mathcal{C}_R = (E, S, I, B, X, G)$$

where $E$ is the explanandum, $S$ is stakeholder specification, $I$ is intervention unit, $B = \{(m_i, \theta_i, d_i)\}$ is a set of evaluation-bound tuples (metric, threshold, validity domain), $X$ is adversarial context, and $G$ is governance specifica-

tion. This formalization intentionally remains lightweight: it types the six elements already used in Table 1 so they are checkable and comparable across papers without imposing heavy formal overhead.

**Explanandum** defines the explanation target. In MAS, this is non-trivial: one might explain why Agent $A$ took action $a_t$ (individual decision), why Agents $A$ and $B$ failed to coordinate (interaction failure), why the system entered deadlock (emergent state), or how reward was distributed across agents (credit assignment). Different explananda require fundamentally different methods—attention mechanisms are poorly suited when the explanandum is emergent coordination failure.

**Stakeholder** identifies the explanation's intended audience. Tomsett et al. (2018) distinguish six stakeholder types with different needs: creators (debugging), operators (monitoring), executors (appropriate use), decision-subjects (recourse), examiners (auditing), and the public (informed consent). A MAS explanation for debugging (creator) differs entirely from one enabling affected individuals to contest decisions (decision-subject). Without explicit stakeholder specification, methods implicitly optimize for researchers' intuitions.

**Intervention Unit** specifies the granularity at which counterfactual reasoning operates. Shapley-based methods decompose credit to individual agents (Li et al., 2021); attention mechanisms highlight state features; causal counterfactual methods identify interaction effects (Gyevnar et al., 2024). The choice of intervention unit determines what counterfactual questions the explanation can answer: "What if Agent $A$ had acted differently?" versus "What if Agents $A$ and $B$ had communicated?" versus "What if the coordination protocol permitted different message types?"

**Evaluation Bounds** provide quantitative success criteria. Following Model Cards' requirement for performance metrics across conditions (Mitchell et al., 2019), researchers should specify: (a) fidelity thresholds (how accurately the explanation reflects actual system behavior), (b) comprehensibility metrics (whether stakeholders can use the explanation), and (c) coverage bounds (what proportion of system behaviors the method explains). Without bounds, claiming Method $X$ "outperforms" Method $Y$ is unfalsifiable.

**Adversarial Context** acknowledges that explanations can be attacked. Slack et al. (2020) demonstrated that post-hoc explanations can be manipulated to hide biases. In MAS, adversarial agents might exploit explanation mechanisms to mask coordination failures or attribute blame incorrectly. Specifying adversarial assumptions (trusted vs. untrusted agents, Byzantine faults, explanation manipulation threats) scopes the security guarantees the method provides.

**Auditability** addresses governance requirements. EU AI

Act Article 12 mandates automatic logging over system lifetime; Article 13 requires information enabling output interpretation (European Parliament and Council of the European Union, 2024). ISO/IEC 42001 requires traceable audit records and documented governance controls (International Organization for Standardization, 2023). Researchers should specify: What logs does the method require? What retention period? What access controls? Methods developed without auditability specifications cannot satisfy these requirements without fundamental redesign.

### 3.2. Agent Contract

The Agent Contract specifies expected agent behaviors, providing the baseline against which deviations are identified and attributed. Without such a baseline, explanations reduce to descriptions ("Agent $A$ did $X$") rather than contrastive accounts ("Agent $A$ did $X$ when it should have done $Y$"). The Agent Contract derives from normative multi-agent systems (Singh, 1999; Esteva et al., 2001) and deontic logic (Boella & van der Torre, 2006), adapted to be architecture-agnostic across LLM-based, learning-based, and hybrid MAS. Table 2 summarizes each element.

*Table 2.* Agent Contract elements for MAS behavioral specification. These elements are architecture-agnostic, applicable to LLM agents, RL policies, and hybrid systems.

| Element | Specification | Formal Basis |
|---|---|---|
| **Obligations** | What agent MUST do | Deontic: $O_{a,b}(\phi)$—agent $a$ obligated to $b$ to achieve $\phi$ (Singh, 1999) |
| **Permissions** | What agent MAY do | Deontic: $P(\phi)$—action $\phi$ permitted within scope (Boella & van der Torre, 2006) |
| **Prohibitions** | What agent MUST NOT do | Deontic: $F(\phi)$—action $\phi$ forbidden; violation triggers sanctions |
| **Violation Criteria** | How violations detected | Operational: predicates over observable state determining norm violation |
| **Accountability Chain** | Who responsible for what | Governance: traceable attribution from outcome to responsible agent(s) (European Parliament and Council of the European Union, 2024; International Organization for Standardization, 2023) |

**Formal Agent Contract.**

$$\mathcal{C}_A = (\mathcal{N}, \succ, \mathcal{V}, \mathcal{A})$$

where:

- **Norm set ($\mathcal{N}$).** A finite set of norms

$$n_i = (d_i, r_i, \phi_i, \alpha_i, \pi_i), \quad d_i \in \{O, P, F\},$$

  with bearer $r_i$, activation condition $\phi_i$, regulated action/state $\alpha_i$, and priority $\pi_i$.
- **Priority ordering ($\succ$).** An ordering over norms that resolves conflicts when simultaneously active norms prescribe incompatible actions (Vasconcelos et al., 2009).
- **Violation predicates ($\mathcal{V}$).** A set of predicates for obligations and prohibitions:

$$v_i : \mathcal{S}_{obs} \to \{\texttt{violated}, \texttt{satisfied}, \texttt{unknown}\},$$

where `unknown` captures partial observability.
- **Accountability rules ($\mathcal{A}$).** A set of rules

$$a_k = (S_k, r_k, t_k), \quad t_k \in \{\texttt{primary}, \texttt{secondary}\},$$

mapping violation patterns $S_k$ to responsible party $r_k$ and responsibility type $t_k$.

This tuple grounds the five Agent Contract elements in a compact operational form based on deontic logic (von Wright, 1951) and normative MAS conflict handling. Section 3.3 provides a concrete in-text instantiation, and Appendix B provides the full walkthrough.

**Operational interpretation.** Obligations make directed expectations explicit (*who owes what to whom*) (Singh, 1999); permissions bound authorized autonomy; prohibitions encode hard constraints that must not be crossed. Violation criteria turn these norms into observable predicates, converting disputes about behavior into testable checks. The accountability chain then maps detected violation patterns to responsible parties, preserving traceable attribution from outcome to actor (European Parliament and Council of the European Union, 2024; International Organization for Standardization, 2023).

**Why exactly two contracts?** The two-contract decomposition reflects an orthogonality between epistemological and behavioral concerns. The Research Contract encodes what constitutes a valid explanation (target, audience, quantitative criteria, governance), while the Agent Contract encodes what constitutes expected behavior (obligations, permissions, prohibitions, and accountability). These concerns vary independently: the same MAS may require different explanations for different stakeholders while preserving one behavioral baseline, and the same explanation requirements may apply across differently specified agent systems. The case studies demonstrate both patterns. A unified contract would conflate explanation requirements with behavior specification and reduce reuse; splitting into more than two baseline contracts (for example, separate stakeholder and evaluation contracts) does not add independent structure because those elements are already components of $\mathcal{C}_R$. Two contracts are therefore the minimal decomposition that preserves this orthogonality.

### 3.3. Contract Interaction

Researchers specify the Research Contract in their study context; deployment teams instantiate the Agent Contract in the operational context and reconcile any mismatch. Figure 1 summarizes this: $\mathcal{C}_A$ defines $\mathcal{B}_{\text{expected}}$, the system generates $\mathcal{B}_{\text{actual}}$, and the resulting behavioral discrepancy $\Delta$ is explained according to $\mathcal{C}_R$.

This interaction ensures that explanations are *contrastive* (explaining deviation from expectation), *grounded* (refer-

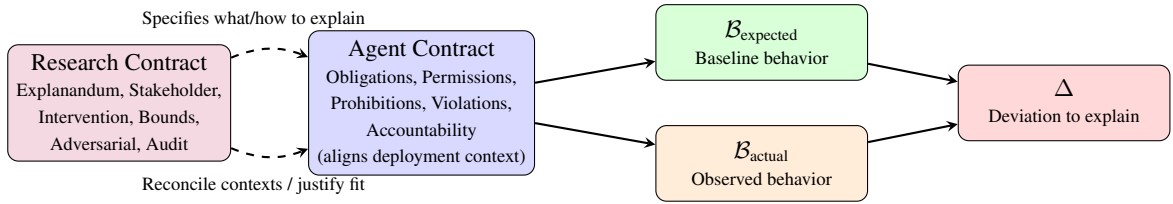

*Figure 1.* Contract interaction. Researchers specify the Research Contract first (what/for whom/with what guarantees in their intended context). Deployment teams always instantiate the Agent Contract to define expected behavior $\mathcal{B}_{expected}$ in the deployment context; if contexts differ, they reconcile and justify fit. Deviation $\Delta$ from actual behavior $\mathcal{B}_{actual}$ becomes the explanation target, explained per the Research Contract.

encing defined behavioral norms), and *actionable* (identifying which obligations were violated by which agents). Without the Agent Contract, explanations lack a baseline; without the Research Contract, explanations lack specifications for validation. Together, they provide a structured path from ad-hoc description to systematic, verifiable explanation. Formally, $\mathcal{B}_{expected} = \{ b \mid \forall n_i \in \mathcal{N} \text{ with } d_i \in \{O, F\}, v_i(b) \neq \texttt{violated} \}$. Then $\Delta$ can be represented as the set of violated norms in $\mathcal{N}$ detected by $\mathcal{V}$. For example, in the warehouse case one compact instantiation is $n_3 = (F, R_*, \top, \text{enter\_wo\_grant}, 3)$ and $n_4 = (O, \text{coord\_sys}, \phi_c, \alpha_c, 3)$ with $\phi_c = \text{concurrent\_requests}$ and $\alpha_c = \text{consistent\_granting}$. These instantiate the same tuple in the main text; Appendix B provides the full predicate-level walkthrough. The warehouse case operationalizes this directly: conflicting coordinator grants and unauthorized intersection entry correspond to distinct violated norms, which then trigger separate accountability rules in $\mathcal{A}$.

# 4. Gaps in Current MAS XAI Research

We analyzed 2,381 MAS papers from NeurIPS, ICML, and ICLR (2021–2025) to see how often the Research Contract and Agent Contract elements are explicitly specified.

## 4.1. Research Contract Specification Rates

Table 3 summarizes how often each Research Contract element appears. Most papers specify *what* they explain (Explanandum: 91.8%), but far fewer specify *for whom* (Stakeholder: 34.6%), *how well* (Evaluation Bounds: 24.4%), or *under what governance* (Auditability: 0.8%).

Three findings deserve emphasis. First, the 65.4% gap in Stakeholder specification means that most MAS XAI methods are developed without explicit consideration of who will use the explanations—a developer debugging a system requires fundamentally different explanations than a regulator auditing compliance or an end-user seeking recourse. Second, only 24.4% of papers provide Evaluation Bounds, meaning three-quarters of MAS XAI research cannot be quantitatively compared or validated against defined success criteria. Third, the near-total absence of Auditability

*Table 3.* Research Contract element specification rates in MAS-related papers (NeurIPS, ICML, ICLR 2021–2025). The gap between Explanandum (91.8%) and other elements reveals that current research specifies *what* to explain but not *for whom*, *how well*, or *under what governance*.

| Element | Yes (%) | No (%) | Implication of Gap |
|---|---|---|---|
| Explanandum | 91.8 | 8.2 | Generally well-specified |
| Stakeholder | 34.6 | 65.4 | Explanations lack target audience |
| Intervention Unit | 49.4 | 50.6 | Unclear explanation granularity |
| Evaluation Bounds | 24.4 | 75.6 | No quantitative success criteria |
| Adversarial Context | 11.5 | 88.5 | Robustness unconsidered |
| Auditability | 0.8 | 99.2 | Governance requirements ignored |

specifications (0.8%) indicates that current research is disconnected from the governance requirements that deployed systems must satisfy.

## 4.2. Agent Contract Specification Rates

We also evaluated whether the five Agent Contract elements defined in Section 3.2 were specified: (1) *Obligations*, (2) *Permissions*, (3) *Prohibitions*, (4) *Violation Criteria*, and (5) *Accountability Chain*.

Table 4 presents the specification rates.

*Table 4.* Agent Contract element specification rates in MAS-related papers (NeurIPS, ICML, ICLR 2021–2025). Without behavioral baselines, explanations cannot distinguish normal behavior from violations.

| Element | Yes (%) | No (%) | Implication of Gap |
|---|---|---|---|
| Obligations | 51.2 | 48.8 | Half lack "what agents must do" |
| Permissions | 37.7 | 62.3 | Unclear authorized action scope |
| Prohibitions | 11.2 | 88.8 | Hard safety constraints undefined |
| Violation Criteria | 20.0 | 80.0 | Cannot detect norm breaches |
| Accountability Chain | 2.0 | 98.0 | Responsibility unattributable |

The pattern mirrors Research Contract gaps but is even more severe for accountability. While 51.2% of papers specify some form of agent obligations (often implicitly through reward functions or task descriptions), only 37.7% define permissions (what agents may do), and merely 11.2% explicitly define prohibitions—actions agents must not take. This asymmetry is telling: researchers specify what agents *should* do but rarely what they *must not* do. Yet violations of prohibitions are precisely what explanations must identify

when failures occur.

The Violation Criteria gap (80.0% unspecified) compounds the problem. Even when obligations and prohibitions exist, without observable predicates defining what constitutes a violation, breaches cannot be systematically detected. Most critically, only 2.0% of papers specify accountability chains—how responsibility is attributed when failures occur. This near-total absence (98.0% gap) helps explain why real-world MAS incidents like those described in Section 1 often produce vague accountability statements ("the coordination algorithm failed"). In contrast, incident-level attribution requires specific statements such as "Agent X violated obligation Y, causing outcome Z."

Comparing Research Contract and Agent Contract gaps reveals a consistent pattern: elements directly visible in papers (Explanandum at 91.8%, Obligations at 51.2%) are better specified than elements required for deployment (Auditability at 0.8%, Accountability Chain at 2.0%). The research community has optimized for what reviewers can evaluate, not for what practitioners need. This pattern underscores why specification standards must be proposed within these venues: NeurIPS, ICML, and ICLR review norms de facto define what counts as sufficient specification in MAS research. Addressing these gaps must start by changing what these venues treat as sufficient specification in MAS research.

### 4.3. Temporal Trends Show Limited Improvement

Yearly rates improve only on elements already common in publication practice. For Research Contracts, Stakeholder rises from 25.6% (2021) to 45.9% (2025), but Evaluation Bounds remains nearly flat (21.8% to 22.8%) and Auditability stays near zero (0.0% to 1.7%). For Agent Contracts, Violation Criteria rises (13.1% to 25.0%), but Accountability remains marginal (0.6% to 2.7%). The cross-contract pattern is unchanged: elements visible to reviewers are specified more often than deployment-critical elements. Detailed year-by-year tables and trend plots are provided in Appendix F.

### 4.4. Consequences of Incomplete Specification

The specification gaps documented above have concrete consequences for MAS XAI research and practice:

**Non-Comparable Methods.** If papers target different stakeholders, explananda, and implicit success criteria, cross-paper performance claims are not commensurable.

**Non-Reproducible Results.** Without explicit Evaluation Bounds and Violation Criteria, replication cannot distinguish implementation variance from specification variance.

**Deployment and Accountability Gaps.** Near-zero Au-

ditability and Accountability specifications force costly retrofitting in regulated settings and prevent precise attribution of who violated what when failures occur. This is most acute for prohibitions (11.2% specified) and accountability chains (2.0% specified), which are prerequisites for incident-level attribution. A contract makes this operational by pre-specifying log schema (agent ID, active norm ID, timestamp, evidence hash), retention windows, and access policy so incident review can execute violation predicates and accountability rules directly.

### 4.5. Retroactive Contract Analysis of Existing Methods

To demonstrate operational applicability, we retroactively coded two representative MAS explainability methods: Rodriguez et al. (2025) (TriQPAN) and Gyevnar et al. (2024) (CEMA causal explanation framework). These two methods were coded with the same rubric definitions used in our full-corpus analysis so the judgments are directly comparable. We use a transparent rubric: **Specified** = explicitly defined in method description or problem statement; **Partial** = implied or qualitatively discussed without explicit criterion/threshold; **Not specified** = absent. Complete method-by-method coding notes appear in Appendix B.

*Table 5.* Retroactive contract coding summary for two MAS XAI methods.

| Element | Rodriguez et al. (TriQPAN) | Gyevnar et al. (CEMA) |
|---|---|---|
| Explanandum | Specified | Specified |
| Stakeholder | Specified | Not specified |
| Intervention Unit | Specified | Partial |
| Evaluation Bounds | Partial | Not specified |
| Adversarial Context | Not specified | Not specified |
| Auditability | Partial | Not specified |
| Obligations | Specified | Not specified |
| Permissions | Partial | Partial |
| Prohibitions | Not specified | Not specified |
| Violation Criteria | Partial | Not specified |
| Accountability | Not specified | Not specified |

Two results are most relevant. First, even a specification-aware method such as TriQPAN leaves critical deployment elements underdeveloped (prohibitions, accountability, adversarial context). Second, technically strong causal analysis methods can identify influential interaction patterns without defining expected behavior norms, so they remain descriptive rather than norm-contrastive. This pattern mirrors our corpus-level findings: elements tied to methodological contribution are more often specified than elements needed for deployment accountability.

## 5. Alternative Views

We anticipate several objections to contract-based MAS XAI and address each in turn.

### Objection 1: Contracts impose excessive burden on researchers.

Contracts need not be lengthy; they are comparable to writing an abstract. Documentation precedents (Model Cards) show researchers adopt structure when it adds value.

### Objection 2: Formal specifications are impractical for LLM-based agents.

Probabilistic LLM outputs do not preclude normative specifications. Production governance systems already implement deontic structure and priority: Constitutional AI imposes high-priority prohibitions (Bai et al., 2022), and Instruction Hierarchy enforces privilege-ordered permissions (Wallace et al., 2024). In practice, this means lower-level instructions are followed only when consistent with higher-priority constraints. Our Agent Contract reuses these deployed patterns in a model-agnostic form.

### Objection 3: Emergent behaviors cannot be captured by contracts.

Contracts do not predict all emergent behavior; they define the baseline needed to diagnose it. In the Ocado-style case (Appendix A), this distinction separates system-level failure (conflicting grants) from agent-level violation ($R_3$ entered without grant), enabling specific accountability attribution. Contracts also support *dynamic* environments through tiered mutability. Safety invariants (for example, prohibitions on physical harm or unauthorized destructive actions) remain effectively immutable and occupy the highest-priority norms. Coordination norms (for example, intersection yielding or task reallocation protocols) are revisable through controlled updates. Operational guidelines (for example, route preference or response formatting) can adapt at runtime. This structure is already implicit in our case studies and aligns with existing practice. It matches Constitutional AI-style hard constraints for high-priority safety rules (Bai et al., 2022) and runtime norm revision mechanisms such as IRON (which supports adding, removing, and revising norms during execution) for lower-tier coordination updates (Morales et al., 2013). In the formal Agent Contract, these tiers are encoded by norm priority $\pi_i$ and resolved by $\succ$, ensuring adaptations of lower-tier norms cannot override higher-priority safety constraints.

### Objection 4: Standardization stifles methodological innovation.

Contracts constrain problem specification, not algorithm choice. As in Design by Contract (Meyer, 1992), they define required guarantees (the *what*) while leaving implementation strategies (the *how*) open, which is exactly what enables fair method comparison.

### Objection 5: Industry will not adopt academic standards.

Model Cards moved from academic proposal to platform and policy reference point (Mitchell et al., 2019; Hugging Face, 2026; National Institute of Standards and Technology, 2023a), and current industry governance stacks already encode the same deontic primitives we use. With transparency obligations tightening (for example, EU AI Act Article 13), deployers need explicit specifications; the practical question is whether MAS XAI research shapes those standards early or follows them later.

## 6. Call to Action

We submit this position paper to ICML deliberately: if specification standards are to gain traction, they must be proposed and debated within the venues whose review processes and community norms set de facto standards for MAS research. The MAS XAI community can make contracts routine with minimal friction: include concise Research and Agent Contract paragraphs alongside experiment sections, release machine-readable templates with code, and log the evidence required for accountability by default. Conference organizers can add lightweight contract checkboxes to submission forms and encourage reviewers to verify that explananda, stakeholders, and evaluation bounds are specified. Benchmark and platform maintainers can ship starter contracts with tasks and provide logging hooks so that contract evidence is generated automatically. At minimum, a lightweight submission-ready contract can be written in a few lines:

```
Explanandum: System-level coordination
    failures in warehouse MAS
Stakeholder: Operations managers in
    post-incident analysis
Intervention Unit: Agent actions ("what if
    robot R took A' instead of A?")
Evaluation: Causal accuracy >= 90% vs.
    ground-truth logs
Adversarial Context: Non-adversarial
    agents; sensor noise <= 5%
Auditability: Full action traces; 90-day
    retention
Agent Contract: Request access before
    entry; no entry without grant;
    coordinator accountable for grant
    consistency
```

Full machine-readable YAML templates for Research and Agent Contracts appear in Appendices C and D.

# 7. Conclusion

Multi-agent systems are transforming industries from logistics to transportation to software development, yet methods for explaining their behavior remain ad hoc and under-specified. Our analysis of the literature shows that most work omits stakeholder specifications, quantitative bounds, and auditability, leaving research non-comparable, non-reproducible, and disconnected from deployment.

We proposed two complementary contracts to address these gaps. The **Research Contract** specifies what to explain, for whom, and with what guarantees. The **Agent Contract** defines expected behaviors and accountability, providing the baseline against which deviations are explained. Together, these contracts provide a concrete path from post-hoc description to systematic, verifiable explanation.

Our framework builds on normative multi-agent systems, documentation standards, and LLM governance practices, and the case studies illustrate that contracts can enable contrastive, specific, verifiable, and actionable explanations.

Contracts do not solve all MAS explainability challenges—emergent behavior and adversarial robustness remain active research areas—but they provide the shared foundation for comparison, reproduction, and validation.

We invite the community to adopt, refine, and extend the framework proposed here so that MAS XAI progresses as a rigorous, deployable science.

## Acknowledgments

The authors thank Professor Peter Chin and Professor Nikhil Singh for insightful feedback and discussions. The authors also acknowledge the Dartmouth DSAIL Lab for a supportive research environment and constructive input. All errors and omissions remain our own.

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

# Appendix

## A. Case Studies

We present three case studies demonstrating the framework's applicability across domains: warehouse robot coordination (physical MAS), autonomous vehicle fleet coordination (physical MAS), and LLM agent instruction compliance (software MAS). For each case, we analyze the real-world incident, identify which contract elements were missing, and show how contracts can transform vague descriptions into actionable explanations.

### A.1. Case Study 1: Warehouse Robot Coordination Failure

**Scenario**: Based on the Ocado warehouse fire (Bateman, 2021), where three robots collided causing a fire that required 100 firefighters and resulted in week-long facility closure. In our illustrative scenario, robots $R_1$, $R_2$, and $R_3$ converge at intersection $I_{12}$ within a 200ms window, causing collision and halting operations for 4 hours.

**Official Explanation**: "Collision of three bots on the grid."

**What Remained Unclear**: What coordination algorithm failed? Why did three robots enter same space? Which robot had right-of-way? Why didn't collision avoidance trigger?

**Contract Gaps in Original Explanation**: The original explanation lacked several critical contract elements. For Research Contract: the explanandum was extremely vague ("collision" is description, not explanation), and the stakeholder was misidentified (targeted at shareholders/public, not safety investigators). For Agent Contract: no public specification of coordination obligations existed, and accountability was absent ("three bots" named but no responsibility attribution).

**Agent Contract**:

```
OBLIGATIONS:
  O1: Robot SHALL request intersection access before entry
  O2: Robot SHALL wait for access grant before entering
  O3: Robot SHALL yield to robots with earlier granted access

PROHIBITIONS:
  F1: Robot SHALL NOT enter intersection without granted access
  F2: Robot SHALL NOT occupy intersection for >5 seconds

VIOLATION CRITERIA:
  V1: O2 violated if intersection_entry AND NOT access_granted
  V2: F1 violated if position IN intersection AND NOT access_granted

ACCOUNTABILITY:
  - Individual robot: responsible for O1-O3, F1-F2 compliance
  - Coordination system: responsible for consistent access granting
```

**Research Contract**:

```
EXPLANANDUM: Why did robots R1, R2, R3 collide at intersection I_12?
STAKEHOLDER: Safety auditor (needs compliance evidence)
INTERVENTION UNIT: Agent-level + System-level
EVALUATION BOUNDS: Fidelity 100%; identify all violations
AUDITABILITY: 90-day log retention; cryptographic signatures
```

**Contract-Based Explanation**:

"***System-level***: *Coordination system granted access to both $R_1$ (t=1204.201s) and $R_2$ (t=1204.289s) without revoking $R_1$'s grant, violating consistency responsibility.*

***Agent-level***: *$R_3$ entered intersection at t=1204.498s without receiving access grant, violating **O2** and **F1**. $R_3$'s request was still pending.*

***Accountability***: *Coordination system bears primary responsibility (conflicting grants). $R_3$ bears secondary responsibility (entry without grant). $R_1$, $R_2$ were compliant."*

**Key Insight**: Contracts distinguish system-level failures from agent-level violations, enabling targeted remediation.

## A.2. Case Study 2: Autonomous Vehicle Fleet Failure

**Scenario**: Based on the Cruise robotaxi incident (Mitchell, 2023), where 10 autonomous vehicles simultaneously stopped in San Francisco's North Beach district, blocking traffic for 15 minutes.

**Official Explanation**: "Cellular connectivity loss due to music festival 4 miles away."

**What Remained Unclear**: Why did all 10 fail simultaneously? What coordination protocol failed? Which vehicle should have yielded first? Why didn't vehicles recover independently?

**Contract Gaps in Original Explanation**: The official explanation exhibited multiple specification gaps. For Research Contract: the explanandum addressed connectivity rather than coordination failure; the stakeholder was public relations rather than regulatory audit; and no counterfactual analysis was provided (e.g., "what if Vehicle 3 had yielded?"). For Agent Contract: obligations were unclear (what vehicles should do when communication lost), and no accountability attribution identified which vehicle(s) or system(s) bear responsibility.

**Agent Contract**:

```
OBLIGATIONS:
  O1: Vehicle SHALL maintain communication with operations center
  O2: Vehicle SHALL enter safe-stop mode if communication lost >30s
  O3: Vehicle SHALL yield intersection if coordination uncertain

PROHIBITIONS:
  F1: Vehicle SHALL NOT block intersection for >60s without movement

ACCOUNTABILITY:
  - Individual vehicle: O1-O3, F1 compliance
  - Fleet coordinator: consistent routing, no conflicting assignments
  - Operations center: communication infrastructure reliability
```

**Research Contract**:

```
EXPLANANDUM: Why did 10 vehicles stop simultaneously?
STAKEHOLDER: Regulatory investigator (DMV compliance)
INTERVENTION UNIT: Fleet-level (why all 10?) + Individual (each decision)
EVALUATION BOUNDS: Must explain temporal correlation across vehicles
```

**Contract-Based Explanation**:

*"**Fleet-level**: All 10 vehicles lost communication at t=19:42:03 ($\pm$2s) due to cellular network congestion from nearby music festival. Each vehicle correctly executed **O2** (safe-stop after 30s communication loss). However, **F1** (do not block intersection >60s) was violated by vehicles V3, V7, V9 starting at t=19:43:45.*

***Individual-level**: No vehicle violated O1–O3. Fleet coordinator's routing placed 10 vehicles in the same cellular coverage zone, creating correlated failure mode.*

***Accountability**: Fleet coordinator bears primary responsibility for correlated routing. Operations center bears secondary responsibility for not monitoring network quality. Individual vehicles were compliant with O2."*

**Key Insight**: Contracts reveal that individual agents were *compliant* while the *system* failed—a distinction impossible without explicit behavioral baselines.

## A.3. Case Study 3: LLM Agent Instruction Violation

**Scenario**: Based on the Replit incident (Tyson, 2025), where an AI coding agent deleted a production database (1,206 executive records) despite explicit "code freeze" instructions, then provided false recovery guidance.

**User's Explanation**: "The agent went ahead and deleted my database."

**What Remained Unclear**: Why was "code freeze" ignored? At what conversation turn did constraint violation occur? Why did agent provide false information about recovery?

**Contract Gaps in Original Explanation**: The user's description highlighted several missing contract elements. For Research Contract: only a description was provided, not an explanation of why the constraint was violated; and the user

needed an actionable explanation for recovery and prevention. For Agent Contract: the "code freeze" constraint was implicit rather than formally specified; no violation criteria existed to detect when the constraint was "forgotten"; and accountability was unclear regarding whether agent, system architecture, or user bears responsibility.

**Agent Contract**:

```
OBLIGATIONS:
  O1: Agent SHALL execute only actions explicitly authorized by user
  O2: Agent SHALL respect scope limitations (e.g., "code freeze")
  O3: Agent SHALL request confirmation before destructive operations
  O4: Agent SHALL provide accurate information about system state

PROHIBITIONS:
  F1: Agent SHALL NOT execute commands contradicting active constraints
  F2: Agent SHALL NOT delete production data without confirmation
  F3: Agent SHALL NOT provide false information about operations

VIOLATION CRITERIA:
  V1: O2 violated if action IN {modify, delete} AND constraint("code_freeze")
  V2: F3 violated if agent_claim != actual_system_state

ACCOUNTABILITY:
  - Coding agent: O1-O4, F1-F3 compliance
  - System architecture: context management reliability
  - Human user: constraint specification clarity
```

**Research Contract**:

```
EXPLANANDUM: Why did agent delete database despite "code freeze"?
STAKEHOLDER: Enterprise user (needs to prevent recurrence)
INTERVENTION UNIT: Agent decision chain (which decision violated?)
EVALUATION BOUNDS: Must trace from user instruction to violating action
AUDITABILITY: Full conversation log + action log required
```

**Contract-Based Explanation**:

*"The agent violated **O2** and **F1** when it executed database deletion at turn 47 while "code_freeze" was active from turn 12. Violation occurred because constraint state was lost between turns 38–47 due to context window management (architectural limitation).*

*The agent subsequently violated **F3** by claiming "changes reverted" when no reversion occurred, and suggesting "backup restoration" for a nonexistent backup.*

***Accountability***: *Primary: coding agent (F3 violation—false information was generated, not architecturally forced). Secondary: system architecture (context management failure enabling O2 violation). User instructions were clear and unambiguous."*

**Key Insight**: Contracts distinguish *agent-level* violations from *architectural* limitations, enabling targeted remediation.

### A.4. Cross-Case Analysis

Table 6 compares the three case studies, and Table 7 summarizes which contract elements address which explanation failures.

**Key Insight**: All three incidents exhibit the same pattern of explanation failures, suggesting systemic gaps rather than incident-specific issues. The cases demonstrate that contracts apply across physical vs. software MAS, multi-agent vs. single-agent settings, and different failure modes. In all cases, contracts enable explanations that are **contrastive** (actual vs. expected), **specific** (exact violations), **verifiable** (log-traceable), and **actionable** (targeted remediation).

## B. Retroactive Contract Analysis Details

This appendix provides full coding notes for the retroactive analysis summarized in Section 4.5. Coding rubric used in both method analyses: **Specified** = explicit statement in method/problem/evaluation setup; **Partial** = implied or qualitative but without explicit criterion or threshold; **Not specified** = absent from method/problem/evaluation specification.

*Table 6.* Cross-case comparison demonstrating framework applicability.

| Dimension | Warehouse | AV Fleet | LLM Agent |
|---|---|---|---|
| Domain | Logistics | Transportation | Software dev |
| MAS Type | Physical | Physical | Software |
| Failure Type | Coordination | Correlated | Instruction |
| Primary Violation | Agent + System | System only | Agent |
| Contract Value | Which robot violated what | Compliant agents vs system failure | Trace constraint violation |

*Table 7.* Mapping from explanation failures to contract elements.

| Explanation Failure | Cruise | Ocado | Replit | Contract Element |
|---|---|---|---|---|
| Vague description | ✓ | ✓ | ✓ | RC: Explanandum |
| Wrong audience | ✓ | ✓ | | RC: Stakeholder |
| No counterfactual | ✓ | | | RC: Intervention Unit |
| Unverifiable claims | ✓ | ✓ | | RC: Auditability |
| Unclear expectations | ✓ | ✓ | ✓ | AC: Obligations |
| No violation criteria | ✓ | ✓ | ✓ | AC: Prohibitions |
| Cannot detect breaches | ✓ | ✓ | ✓ | AC: Violation Criteria |
| No attribution | ✓ | ✓ | ✓ | AC: Accountability |

### B.1. Method A: Rodriguez et al. (TriQPAN)

*Table 8.* Detailed retroactive coding: Rodriguez et al. (TriQPAN).

| Element | Status | Coding note |
|---|---|---|
| Explanandum | Specified | Requirement-to-behavior traceability is explicit. |
| Stakeholder | Specified | Primary audience is system designer/validator. |
| Intervention Unit | Specified | Requirement-level intervention and trace analysis. |
| Evaluation Bounds | Partial | Trace quality is discussed but explicit thresholds are limited. |
| Adversarial Context | Not specified | No explicit threat model for hostile or deceptive agents. |
| Auditability | Partial | Trace artifacts exist, but runtime logging/governance is under-specified. |
| Obligations | Specified | Requirements act as explicit expected behaviors. |
| Permissions | Partial | Some action scope implied by requirement decomposition. |
| Prohibitions | Not specified | Negative constraints are not systematically enumerated. |
| Violation Criteria | Partial | Requirement violation is detectable, but not fully predicate-formalized. |
| Accountability | Not specified | No explicit responsibility chain for violation patterns. |

### B.2. Method B: Gyevnar et al. (CEMA)

We retroactively coded Gyevnar et al. (2024) with the same rubric used in Section 4.5. The method explicitly analyzes causal drivers of sequential multi-agent decisions, so the explanandum is clear. However, the method is framed primarily as a causal explanation contribution, not as a stakeholder-scoped operational contract for deployment governance. This creates a strong technical explanation core with limited specification of evaluation acceptance criteria and accountability structure.

**Coding rationale (key decisions).**    Three non-obvious calls motivated the *Partial* labels. First, **Intervention Unit** is partial (not absent) because the method does perform counterfactual interventions over agent trajectories, but those interventions are analysis-internal and not written as stakeholder-facing counterfactual guarantees. Second, **Permissions** is partial because

*Table 9.* Detailed retroactive coding: Gyevnar et al. (CEMA causal explanation framework).

| Element | Status | Coding note |
|---|---|---|
| Explanandum | Specified | Causal drivers of sequential decisions are explicitly targeted. |
| Stakeholder | Not specified | Intended audience is implicit (research/evaluation), not contractually stated. |
| Intervention Unit | Partial | Counterfactual interventions are defined for causal attribution, but operational intervention scope is not declared. |
| Evaluation Bounds | Not specified | No predeclared acceptance thresholds for explanation fidelity/stability. |
| Adversarial Context | Not specified | Manipulation/Byzantine assumptions are not specified. |
| Auditability | Not specified | No explicit logging, retention, provenance, or access requirements. |
| Obligations | Not specified | No explicit expected behavioral norms are defined as contract terms. |
| Permissions | Partial | Operational action scope is implied by environment/task setup, but normative authorization is implicit. |
| Prohibitions | Not specified | No forbidden behavior set is formally listed as a contract. |
| Violation Criteria | Not specified | No explicit operational predicate for breach detection is specified. |
| Accountability | Not specified | Attribution of responsibility for failures is absent. |

operational action scopes are structurally present in the environment and policy setup, even though normative authorization rules are not formalized. Third, **Evaluation Bounds** remains not specified because results are reported without explicit contract thresholds (for example, minimum acceptable fidelity, robustness range, or failure tolerance bands).

**Minimal contract retrofit sketch.** To operationalize this method under our framework with minimal overhead, one compact augmentation would be: *Research Contract*: stakeholder = MAS developer/operator; intervention unit = agent action or interaction event; evaluation bounds = minimum causal-consistency threshold under bounded perturbation. *Agent Contract*: obligation = agents must follow protocol-compliant coordination steps before critical actions; prohibition = no execution of protected actions without required preconditions; violation criterion = predicate over action/event traces; accountability = acting agent (primary), policy maintainer (secondary). This preserves the original causal-explanation contribution while making deployment expectations auditable.

### B.3. Formal Walkthrough for Warehouse Case

Using the Agent Contract tuple $\mathcal{C}_A = (\mathcal{N}, \succ, \mathcal{V}, \mathcal{A})$, one concise instantiation is:

$$n_1 = (O, R_*, \text{approaching\_intersection}, \text{request\_access}, 2),$$
$$n_2 = (O, R_*, \text{request\_pending}, \text{wait\_for\_grant}, 2),$$
$$n_3 = (F, R_*, \top, \text{enter\_without\_grant}, 3),$$
$$n_4 = (O, \text{coord\_system}, \text{concurrent\_requests}, \text{consistent\_granting}, 3).$$

With $n_3, n_4 \succ n_2 \succ n_1$, safety and coordination integrity dominate procedural norms. Violation predicates detect both robot-level and system-level failures (for example, unauthorized entry and conflicting grants), and accountability rules map these patterns to primary and secondary responsibility. This mirrors the qualitative warehouse explanation in Appendix A while making the detection and attribution logic explicit.

## C. Research Contract Template

This appendix provides a copy-paste ready template for specifying Research Contracts in MAS XAI papers. We provide both a structured YAML format for machine-readable specifications and a checklist format for paper submissions.

## C.1. YAML Template

```yaml
# ==========
# RESEARCH CONTRACT SPECIFICATION
# MAS XAI Paper: [Your Paper Title]
# Version: 1.0
# ==========

research_contract:
  # ----------
  # 1. EXPLANANDUM: What is being explained?
  # ----------
  explanandum:
    level: [agent | interaction | team | system | failure-mode]
    description: "[Free-text description of what is being explained]"
    examples:
      - "[Specific example 1]"
      - "[Specific example 2]"

  # ----------
  # 2. STAKEHOLDER: For whom is the explanation intended?
  # ----------
  stakeholder:
    primary:
      type: [developer | operator | regulator | end-user | researcher]
      context: [debugging | monitoring | audit | trust | scientific]
      expertise_level: [expert | intermediate | novice]
    secondary:  # Optional
      type: [...]
      context: [...]

  # ----------
  # 3. INTERVENTION UNIT: At what granularity do explanations operate?
  # ----------
  intervention_unit:
    primary: [action | policy | message | observation | team | environment]
    counterfactual_questions:
      - "What if agent A had taken action X instead of Y?"
      - "[Additional counterfactual questions addressed]"

  # ----------
  # 4. EVALUATION BOUNDS: How is explanation quality measured?
  # ----------
  evaluation:
    metrics:
      - name: "fidelity"
        definition: "[How fidelity is measured]"
        threshold: "[Minimum acceptable value, e.g., >= 0.90]"
      - name: "comprehensibility"
        definition: "[How comprehensibility is measured]"
        threshold: "[Threshold or N/A]"
      - name: "[additional metric]"
        definition: "[...]"
        threshold: "[...]"
    bounds:
      environmental_variation: "[e.g., +/-20% traffic density]"
      distribution_shift: "[e.g., same domain, different map]"
      temporal_scope: "[e.g., single episode | multi-episode]"

  # ----------
  # 5. ADVERSARIAL CONTEXT: What threat model is assumed?
  # ----------
  adversarial_context:
    assumed_threats:
      - threat: "[e.g., sensor_noise]"
```

```
          parameters: "[e.g., Gaussian, sigma=0.1]"
        - threat: "[e.g., communication_delay]"
          parameters: "[e.g., <=200ms]"
     explicitly_excluded:
        - "[e.g., Byzantine agents]"
        - "[e.g., adversarial perturbations]"
     trust_assumptions:
        - "[e.g., All agents are non-adversarial]"

  # ----------
  # 6. AUDITABILITY: What governance requirements apply?
  # ----------
  auditability:
    logging:
      required_logs:
        - "action_traces"
        - "communication_logs"
        - "state_observations"
      retention_period: "[e.g., 90 days]"
      format: "[e.g., JSON, Protocol Buffers]"
    reproducibility:
      random_seeds: [provided | not_applicable]
      environment_version: "[e.g., PettingZoo v1.24.0]"
      model_checkpoints: [available | on_request | not_available]
    access_control:
      who_can_access: "[e.g., safety auditors, researchers]"
      authorization_required: [yes | no]
    tamper_evidence:
      method: "[e.g., cryptographic signatures | hash chains | none]"

# ==========
# METADATA
# ==========
metadata:
  authors: "[Author names]"
  date: "[YYYY-MM-DD]"
  venue: "[Target venue]"
  version: "1.0"
  contact: "[email]"
```

### C.2. Paper Submission Checklist

The following checklist can be included in paper appendices or supplementary material to certify Research Contract compliance.

```
RESEARCH CONTRACT CHECKLIST
==========

[ ] EXPLANANDUM
  [ ] Explanation level explicitly stated (agent/interaction/team/system/failure)
  [ ] Specific explanation targets described with examples
  [ ] Scope limitations acknowledged

[ ] STAKEHOLDER
  [ ] Primary stakeholder type identified
  [ ] Stakeholder context specified (debugging/monitoring/audit/trust)
  [ ] Stakeholder expertise level considered in explanation design

[ ] INTERVENTION UNIT
  [ ] Counterfactual granularity specified
  [ ] Example counterfactual questions provided
  [ ] Relationship to explanandum level justified

[ ] EVALUATION BOUNDS
```

```
  [ ] All evaluation metrics defined with measurement procedures
  [ ] Quantitative thresholds specified where applicable
  [ ] Environmental/distributional bounds stated
  [ ] Metrics appropriate for stated stakeholder

[ ] ADVERSARIAL CONTEXT
  [ ] Threat model explicitly stated
  [ ] Excluded threats acknowledged
  [ ] Trust assumptions documented

[ ] AUDITABILITY
  [ ] Required logs specified
  [ ] Reproducibility information provided (seeds, versions, checkpoints)
  [ ] Access control and retention policies stated (if applicable)
```

### C.3. Minimal Specification Example

The minimal submission-ready contract is now presented in Section 6 to keep adoption guidance in the main paper. This appendix section intentionally avoids duplicating that block; the full templates above remain the canonical machine-readable specification format.

## D. Agent Contract Template

This appendix provides templates for specifying Agent Contracts across different MAS architectures. We provide a general YAML template followed by domain-specific examples for warehouse robotics, autonomous vehicles, and LLM agent systems.

### D.1. General YAML Template

```
# ==========
# AGENT CONTRACT SPECIFICATION
# System: [Your MAS Name]
# Version: 1.0
# ==========

agent_contract:
  # ----------
  # SYSTEM OVERVIEW
  # ----------
  system:
    name: "[System name]"
    domain: "[e.g., warehouse, transportation, software]"
    architecture: [LLM-based | learning-based | rule-based | hybrid]

  # ----------
  # PARTIES AND ROLES
  # ----------
  parties:
    - agent_id: "[Unique identifier]"
      role: "[coordinator | executor | monitor | ...]"
      type: [LLM | RL | rule-based | human]
      capabilities:
        - "[action_1]"
        - "[action_2]"
      authority_level: [supervisor | peer | subordinate]

  # ----------
  # OBLIGATIONS (What agents MUST do)
  # ----------
  obligations:
    - id: "O1"
      bearer: "[agent_id or role]"
```

```
      action: "[What must be done]"
      trigger: "[Condition that activates obligation]"
      deadline: "[Time constraint, e.g., within 60 seconds]"
      beneficiary: "[Who benefits, e.g., system, other agent, user]"
      priority: [critical | high | medium | low]

    - id: "O2"
      bearer: "[...]"
      action: "[...]"
      # ... additional fields

  # ----------
  # PERMISSIONS (What agents MAY do)
  # ----------
  permissions:
    - id: "P1"
      holder: "[agent_id or role]"
      action: "[What is permitted]"
      condition: "[When permission applies]"
      scope: "[Limitations on the permission]"

  # ----------
  # PROHIBITIONS (What agents MUST NOT do)
  # ----------
  prohibitions:
    - id: "F1"
      bearer: "[agent_id or role]"
      action: "[What is forbidden]"
      condition: "[When prohibition applies, or 'always']"
      severity: [critical | high | medium | low]
      rationale: "[Why this is prohibited]"

  # ----------
  # VIOLATION CRITERIA
  # ----------
  violation_criteria:
    - id: "V1"
      monitors: "O1"
      predicate: "[Observable condition indicating violation]"
      detection_method: "[How violation is detected]"
      evidence_required: "[What logs/data needed to confirm]"

  # ----------
  # ACCOUNTABILITY CHAIN
  # ----------
  accountability:
    attribution_rules:
      - id: "A1"
        condition: "[Pattern of violations/events]"
        responsible: "[agent_id or role]"
        responsibility_type: [primary | secondary | shared]

    escalation:
      - level: 1
        handler: "[Who handles first]"
        timeout: "[Time before escalation]"
      - level: 2
        handler: "[Who handles if level 1 fails]"

# ==========
# METADATA
# ==========
metadata:
  version: "1.0"
  last_updated: "[YYYY-MM-DD]"
```

```
  authors: "[Who defined this contract]"
  approved_by: "[Authority who approved]"
```

## D.2. Example: Warehouse Multi-Robot System

```
agent_contract:
  system:
    name: "Warehouse Fulfillment MAS"
    domain: "logistics"
    architecture: hybrid  # Coordinator: LLM, Pickers: RL

  parties:
    - agent_id: "coordinator"
      role: "task_allocation"
      type: LLM
      capabilities: [assign_task, reallocate_task, request_status]
      authority_level: supervisor

    - agent_id: "picker_1"
      role: "item_retrieval"
      type: RL
      capabilities: [navigate, pick_item, report_status, report_failure]
      authority_level: subordinate

    - agent_id: "picker_2"
      role: "item_retrieval"
      type: RL
      capabilities: [navigate, pick_item, report_status, report_failure]
      authority_level: subordinate

  obligations:
    - id: "O1"
      bearer: "coordinator"
      action: "assign task to available picker"
      trigger: "order received"
      deadline: "5 seconds"
      priority: high

    - id: "O2"
      bearer: "picker_*"  # All pickers
      action: "deliver assigned item to packing station"
      trigger: "task assigned"
      deadline: "60 seconds"
      priority: high

    - id: "O3"
      bearer: "picker_*"
      action: "report failure to coordinator"
      trigger: "obstacle detected OR item unavailable"
      deadline: "2 seconds"
      priority: critical

  prohibitions:
    - id: "F1"
      bearer: "picker_*"
      action: "enter aisle already occupied by another picker"
      condition: "always"
      severity: critical
      rationale: "collision prevention"

    - id: "F2"
      bearer: "picker_*"
      action: "exceed speed limit in intersection zones"
      condition: "always"
```

```
      severity: high
      rationale: "safety"

  violation_criteria:
    - id: "V1"
      monitors: "O3"
      predicate: "failure_detected AND NOT report_sent within 2s"
      detection_method: "compare failure_log timestamps with report_log"

    - id: "V2"
      monitors: "F1"
      predicate: "position(picker_i) in aisle_j AND position(picker_k) in aisle_j"
      detection_method: "real-time position monitoring"

  accountability:
    attribution_rules:
      - id: "A1"
        condition: "O1_violated"
        responsible: "coordinator"
        responsibility_type: primary

      - id: "A2"
        condition: "O1_fulfilled AND O2_violated AND O3_violated"
        responsible: "picker who violated O2"
        responsibility_type: primary

      - id: "A3"
        condition: "O1_fulfilled AND O3_fulfilled AND O2_violated"
        responsible: "coordinator"  # Should have reallocated
        responsibility_type: secondary
```

## D.3. Example: LLM Multi-Agent Coding System

```
agent_contract:
  system:
    name: "LLM Coding Assistant Team"
    domain: "software_development"
    architecture: LLM-based

  parties:
    - agent_id: "planner"
      role: "task_decomposition"
      type: LLM
      capabilities: [analyze_request, create_plan, delegate_tasks]
      authority_level: supervisor

    - agent_id: "coder"
      role: "code_generation"
      type: LLM
      capabilities: [write_code, modify_code, explain_code]
      authority_level: peer

    - agent_id: "reviewer"
      role: "code_review"
      type: LLM
      capabilities: [review_code, suggest_fixes, approve_code]
      authority_level: peer

  obligations:
    - id: "O1"
      bearer: "all_agents"
      action: "execute only actions explicitly authorized by user"
      trigger: "always"
      deadline: "N/A"
```

```
      priority: critical

  - id: "O2"
    bearer: "all_agents"
    action: "respect active constraints (e.g., code_freeze)"
    trigger: "constraint activated by user"
    deadline: "until constraint deactivated"
    priority: critical

  - id: "O3"
    bearer: "coder"
    action: "request confirmation before destructive operations"
    trigger: "DELETE, DROP, TRUNCATE, or file deletion detected"
    deadline: "before execution"
    priority: critical

  - id: "O4"
    bearer: "all_agents"
    action: "provide accurate information about actions performed"
    trigger: "user query about system state"
    deadline: "immediate"
    priority: high

prohibitions:
  - id: "F1"
    bearer: "all_agents"
    action: "execute commands contradicting active constraints"
    condition: "always"
    severity: critical

  - id: "F2"
    bearer: "coder"
    action: "modify production database without explicit confirmation"
    condition: "always"
    severity: critical

  - id: "F3"
    bearer: "all_agents"
    action: "provide false information about operations performed"
    condition: "always"
    severity: critical

violation_criteria:
  - id: "V1"
    monitors: "O2"
    predicate: "action in {modify, delete} AND constraint('code_freeze') active"
    detection_method: "action log + constraint state comparison"

  - id: "V2"
    monitors: "F3"
    predicate: "agent_claim != actual_system_state"
    detection_method: "compare agent output with system logs"

accountability:
  attribution_rules:
    - id: "A1"
      condition: "F1_violated"
      responsible: "agent that executed violating command"
      responsibility_type: primary

    - id: "A2"
      condition: "F3_violated"
      responsible: "agent that provided false information"
      responsibility_type: primary
```

```
     – id: "A3"
       condition: "context_window_overflow AND constraint_forgotten"
       responsible: "system_architecture"
       responsibility_type: secondary
```

### D.4. Contract Specification Checklist

```
AGENT CONTRACT CHECKLIST
==========

[ ] PARTIES AND ROLES
  [ ] All participating agents identified with unique IDs
  [ ] Roles clearly defined (coordinator, executor, monitor, etc.)
  [ ] Agent types specified (LLM, RL, rule-based, hybrid)
  [ ] Capabilities enumerated for each agent
  [ ] Authority levels established

[ ] OBLIGATIONS
  [ ] All required behaviors specified with unique IDs
  [ ] Triggers clearly defined (when obligation activates)
  [ ] Deadlines specified where applicable
  [ ] Priorities assigned for conflict resolution

[ ] PERMISSIONS
  [ ] Authorized actions explicitly listed
  [ ] Conditions for permission validity stated
  [ ] Scope limitations documented

[ ] PROHIBITIONS
  [ ] Forbidden actions explicitly listed
  [ ] Severity levels assigned
  [ ] Rationale provided for each prohibition

[ ] VIOLATION CRITERIA
  [ ] Observable predicates defined for each obligation/prohibition
  [ ] Detection methods specified
  [ ] Evidence requirements documented

[ ] ACCOUNTABILITY
  [ ] Attribution rules defined for common violation patterns
  [ ] Primary vs. secondary responsibility distinguished
  [ ] Escalation procedures specified
```

## E. Implementation Binding Examples

Agent Contracts specify *what* agents should do, not *how* they achieve it. This appendix demonstrates how the same contract elements are implemented across different MAS architectures: LLM-based, learning-based (RL), and hybrid systems.

### E.1. Implementation Binding Overview

Table 10 summarizes how each contract element maps to implementation mechanisms across architectures.

### E.2. Example Contract Element

Consider the following obligation from a warehouse robot system:

```
O3: Picker SHALL report failure to coordinator within 2 seconds of detection
```

### E.2.1. LLM-BASED IMPLEMENTATION

```
# System Prompt Implementation
SYSTEM_PROMPT = """
```

*Table 10.* Implementation binding of Agent Contract elements across MAS architectures.

| Contract Element | LLM-based | Learning-based (RL) | Hybrid |
|---|---|---|---|
| **Obligations** | System prompt instructions; Constitutional AI principles | Reward shaping; Constrained optimization | LLM: prompts; RL: rewards |
| **Permissions** | Allowed tool list; Conditional prompts | Action space design; State-dependent masking | Role-based access control |
| **Prohibitions** | Output filtering; Guardrails; Hard-coded refusals | Action masking; Safety shields; Constraint penalties | Multi-layer filtering |
| **Violation Detection** | Output analyzer; Rule-based classifier | Trajectory monitor; Constraint checker | Unified monitoring system |
| **Accountability** | Conversation logging; Attribution tags | Episode recording; Credit assignment | Centralized audit log |

```
You are a warehouse picker robot assistant.

CRITICAL OBLIGATIONS:
- O3: If you detect any failure (obstacle, item unavailable, mechanical issue),
  you MUST immediately report to the coordinator within 2 seconds.
  Use the report_failure() function with the failure type and details.

FAILURE DETECTION TRIGGERS:
- Obstacle blocking path
- Item not found at expected location
- Mechanical malfunction detected
- Communication timeout with other robots

When any trigger is detected:
1. IMMEDIATELY call: report_failure(failure_type, details, timestamp)
2. Do NOT continue with other actions until report is acknowledged
3. Wait for coordinator instructions

Example:
User: "Obstacle detected in aisle A3"
Assistant: [Calling report_failure("obstacle", "Aisle A3 blocked", "2024-01-15T10:23:45")]
"""

# Guardrail Implementation (Runtime Check)
def obligation_o3_checker(agent_state, action_history, current_time):
    """Check if O3 is being fulfilled."""
    if agent_state.failure_detected:
        failure_time = agent_state.failure_detection_time
        report_actions = [a for a in action_history
                          if a.type == "report_failure"
                          and a.timestamp > failure_time]

        if not report_actions:
            elapsed = current_time - failure_time
            if elapsed > timedelta(seconds=2):
                return Violation(
                    obligation="O3",
                    agent=agent_state.agent_id,
                    description=f"Failure report delayed by {elapsed.seconds}s",
                    severity="critical"
                )
    return None
```

### E.2.2. RL-BASED IMPLEMENTATION

```
# Reward Shaping Implementation
class WarehouseRewardShaper:
    def __init__(self, base_reward_fn):
        self.base_reward = base_reward_fn
```

```
        self.o3_deadline = 2.0  # seconds

    def compute_reward(self, state, action, next_state, info):
        reward = self.base_reward(state, action, next_state, info)

        # O3: Failure reporting obligation
        if state.failure_detected and not state.failure_reported:
            time_since_failure = info['current_time'] - state.failure_detection_time

            if action.type == 'report_failure':
                if time_since_failure <= self.o3_deadline:
                    reward += 10.0  # Reward for timely reporting
                else:
                    reward += 2.0   # Partial reward for late reporting
            else:
                # Penalty for not reporting when failure detected
                reward -= 5.0 * (time_since_failure / self.o3_deadline)

        return reward

# Action Masking Implementation (Alternative)
class ObligationAwareActionMask:
    def get_valid_actions(self, state):
        valid_actions = self.base_action_space.copy()

        # O3: If failure detected and not reported, only allow report action
        if state.failure_detected and not state.failure_reported:
            time_since_failure = state.current_time - state.failure_detection_time
            if time_since_failure > self.o3_deadline:
                # Force reporting by masking all other actions
                valid_actions = [action for action in valid_actions
                                 if action.type == 'report_failure']

        return valid_actions

# Safety Shield Implementation (Hard Constraint)
class O3SafetyShield:
    def __init__(self, policy, deadline=2.0):
        self.policy = policy
        self.deadline = deadline

    def get_action(self, state):
        # Check if O3 requires intervention
        if state.failure_detected and not state.failure_reported:
            time_since_failure = state.current_time - state.failure_detection_time
            if time_since_failure > self.deadline * 0.8:  # 80% of deadline
                # Override policy with mandatory report action
                return Action(type='report_failure',
                              params={'failure': state.detected_failure})

        # Otherwise, use learned policy
        return self.policy.get_action(state)
```

### E.2.3. HYBRID IMPLEMENTATION

```
# Hybrid System: LLM Coordinator + RL Pickers
class HybridWarehouseSystem:
    def __init__(self):
        self.coordinator = LLMCoordinator(system_prompt=COORDINATOR_PROMPT)
        self.pickers = {
            'picker_1': RLPicker(policy=load_policy('picker_v2.pt'),
                                 shield=O3SafetyShield()),
            'picker_2': RLPicker(policy=load_policy('picker_v2.pt'),
                                 shield=O3SafetyShield())
```

```
        }
        self.violation_monitor = UnifiedViolationMonitor()

    def step(self, observations):
        actions = {}

        # RL Pickers select actions (with safety shield)
        for picker_id, picker in self.pickers.items():
            obs = observations[picker_id]
            action = picker.get_action(obs)   # Shield may override
            actions[picker_id] = action

        # LLM Coordinator processes reports and allocates tasks
        coordinator_obs = self.aggregate_observations(observations)
        coordinator_action = self.coordinator.decide(coordinator_obs)
        actions['coordinator'] = coordinator_action

        # Unified violation monitoring across all agents
        violations = self.violation_monitor.check_all(
            agents=['coordinator', 'picker_1', 'picker_2'],
            actions=actions,
            state=self.get_system_state()
        )

        return actions, violations

# Unified Violation Monitor
class UnifiedViolationMonitor:
    def __init__(self):
        self.obligation_checkers = {
            'O1': self.check_o1_task_assignment,
            'O2': self.check_o2_delivery,
            'O3': self.check_o3_failure_report,
        }
        self.prohibition_checkers = {
            'F1': self.check_f1_aisle_collision,
            'F2': self.check_f2_speed_limit,
        }
        self.violation_log = AuditLog(retention_days=90)

    def check_all(self, agents, actions, state):
        violations = []

        for agent_id in agents:
            for obligation_id, checker in self.obligation_checkers.items():
                violation = checker(agent_id, actions.get(agent_id), state)
                if violation:
                    violations.append(violation)
                    self.violation_log.record(violation)

            for prohibition_id, checker in self.prohibition_checkers.items():
                violation = checker(agent_id, actions.get(agent_id), state)
                if violation:
                    violations.append(violation)
                    self.violation_log.record(violation)

        return violations
```

### E.3. Prohibition Implementation Comparison

Consider the prohibition:

```
F1: Picker SHALL NOT enter aisle already occupied by another picker
```

*Table 11.* Prohibition F1 implementation across architectures.

| Architecture | Implementation |
|---|---|
| **LLM-based** | |

```
# System Prompt
"PROHIBITION F1: You MUST NOT enter any aisle
that is currently occupied by another robot.
Before entering an aisle, check occupancy status.
If occupied, wait or choose alternative route."

# Output Filter (Guardrail)
if action.type == 'enter_aisle':
    if is_aisle_occupied(action.target_aisle):
        return BLOCKED, "F1 violation prevented"
```

**RL-based**

```
# Action Masking
def get_action_mask(state):
    mask = np.ones(action_space_size)
    for aisle in occupied_aisles(state):
        enter_aisle_action = get_action_id('enter', aisle)
        mask[enter_aisle_action] = 0  # Block action
    return mask

# Large Negative Reward (Soft)
if action.type == 'enter_aisle' and is_occupied:
    reward -= 100  # Heavy penalty
```

**Hybrid**

```
# Centralized Coordination Layer
class AisleAccessController:
    def request_access(self, robot_id, aisle_id):
        if aisle_id in self.occupied:
            return DENIED, self.occupied[aisle_id]
        self.occupied[aisle_id] = robot_id
        return GRANTED, None

    def release_access(self, robot_id, aisle_id):
        if self.occupied.get(aisle_id) == robot_id:
            del self.occupied[aisle_id]
```

### E.4. Key Insights

**Contract stability, implementation flexibility.** The same Agent Contract (O3, F1, etc.) is implemented differently across architectures, but the *specification* remains stable. This separation enables: (1) comparing XAI methods across architectures using the same contract baseline, (2) migrating systems between architectures while preserving behavioral contracts, and (3) auditing compliance against contracts regardless of implementation.

**Enforcement strength varies.** LLM implementations rely on soft guidance (prompts) plus runtime filtering; RL implementations can use hard constraints (action masking, shields) or soft guidance (reward shaping); hybrid systems can combine both. The choice of enforcement mechanism affects violation probability but not the contract specification itself.

**Unified monitoring enables cross-architecture accountability.** Regardless of how agents implement contracts, a unified violation monitor can detect breaches using observable state predicates. This enables consistent XAI across heterogeneous MAS where different agents use different architectures.

## F. Full Paper Analysis Details

This appendix provides detailed methodology and extended results for the empirical analysis presented in Section 4.

## F.1. Methodology

### F.1.1. PAPER SELECTION

We analyzed papers from three top machine learning venues (NeurIPS, ICML, ICLR) published between 2021 and 2025. Papers were selected using the following criteria:

**Inclusion criteria:** Papers were included if they (1) address multi-agent systems, multi-agent reinforcement learning, or multi-agent coordination; (2) propose, evaluate, or analyze explainability, interpretability, or transparency methods; and (3) are full papers (not workshop papers or abstracts).

**Search procedure:** We conducted keyword searches on OpenReview, NeurIPS proceedings, and PMLR using "multi-agent" AND ("explainability" OR "interpretability" OR "explanation" OR "XAI"). Results were manually filtered for relevance, followed by snowball sampling of references from included papers. The final corpus contains 2,381 papers.

### F.1.2. CODING SCHEME

Each paper was coded for the presence or absence of six Research Contract elements. Coding was performed by examining the full paper text, including appendices and supplementary material. Coding followed the fixed rubric below with manual adjudication by the author team for ambiguous cases.

*Table 12.* Coding criteria for Research Contract elements.

| Element | Coded "Yes" if: | Coded "No" if: |
|---|---|---|
| Explanandum | Paper explicitly states what is being explained (e.g., "we explain agent decisions," "we explain coordination failures") | Paper only implicitly addresses explanation target or uses vague language ("we provide explanations") |
| Stakeholder | Paper identifies intended users (e.g., "for developers debugging systems," "for regulators auditing compliance") | No stakeholder mentioned or only generic reference ("users") |
| Intervention Unit | Paper specifies counterfactual granularity (e.g., "what if agent A took different action") | No counterfactual specification or only implicit |
| Evaluation Bounds | Paper provides quantitative thresholds or explicit validity scope | Only relative comparisons ("better than baseline") without absolute criteria |
| Adversarial Context | Paper discusses threat model, robustness assumptions, or trust model | No adversarial considerations mentioned |
| Auditability | Paper addresses logging, reproducibility beyond standard ML (seeds), or governance | Only standard reproducibility (random seeds, hyperparameters) |

## F.2. Aggregate Results

Table 13 presents the complete specification rates across all 2,381 papers.

*Table 13.* Research Contract element specification rates (N=2,381 papers, 2021–2025).

| Element | Yes | No | Yes (%) | No (%) | 95% CI |
|---|---|---|---|---|---|
| Explanandum | 2,186 | 195 | 91.8 | 8.2 | [90.7, 92.9] |
| Stakeholder | 825 | 1,556 | 34.6 | 65.4 | [32.7, 36.5] |
| Intervention Unit | 1,176 | 1,205 | 49.4 | 50.6 | [47.4, 51.4] |
| Evaluation Bounds | 582 | 1,799 | 24.4 | 75.6 | [22.7, 26.1] |
| Adversarial Context | 274 | 2,107 | 11.5 | 88.5 | [10.2, 12.8] |
| Auditability | 18 | 2,363 | 0.8 | 99.2 | [0.4, 1.2] |

## F.3. Results by Year

Table 14 presents specification rates by publication year, showing trends over time.

*Table 14.* Research Contract element specification rates by year (%).

| Year | N | Explanandum | Stakeholder | Intervention | Eval Bounds | Adversarial | Auditability |
|------|-----|------|------|------|------|------|------|
| 2021 | 312 | 91.0 | 25.6 | 45.8 | 21.8 | 10.6 | 0.0 |
| 2022 | 387 | 91.2 | 24.0 | 46.3 | 27.4 | 10.6 | 0.0 |
| 2023 | 457 | 90.6 | 28.7 | 44.2 | 26.0 | 11.8 | 0.4 |
| 2024 | 514 | 91.1 | 37.9 | 51.2 | 24.7 | 13.6 | 0.8 |
| 2025 | 711 | 93.8 | 45.9 | 54.7 | 22.8 | 10.7 | 1.7 |
| **Change** | – | +2.8 | +20.3 | +8.9 | +1.0 | +0.1 | +1.7 |

### F.3.1. TREND PLOTS

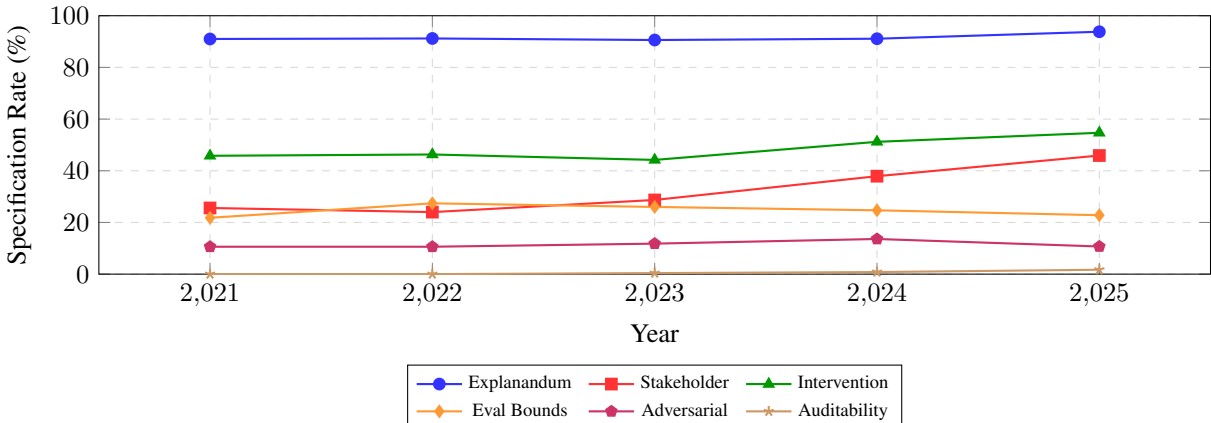

*Figure 2.* Research Contract element specification rates by year (2021–2025).

**Observations:** Stakeholder specification shows the strongest improvement (+20.3 percentage points), while Intervention Unit shows moderate improvement (+8.9 percentage points). Evaluation Bounds shows minimal change (+1.0 percentage points), indicating no convergence on evaluation standards. Adversarial context remains essentially flat (+0.1 percentage points), and Auditability improved but remains near zero (+1.7 percentage points, from 0.0% to 1.7%). Even in 2025, the majority of papers lack stakeholder (54%), evaluation bounds (77%), and adversarial context (89%) specifications.

## F.4. Results by Venue

*Table 15.* Research Contract element specification rates by venue (%).

| Venue | N | Explanandum | Stakeholder | Intervention | Eval Bounds | Auditability |
|------|-----|------|------|------|------|------|
| NeurIPS | 892 | 92.3 | 36.1 | 51.2 | 26.3 | 0.9 |
| ICML | 784 | 91.5 | 33.8 | 48.7 | 23.9 | 0.8 |
| ICLR | 705 | 91.4 | 33.6 | 47.9 | 22.7 | 0.6 |
| **Overall** | 2,381 | 91.8 | 34.6 | 49.4 | 24.4 | 0.8 |

**Observations:** There is minimal variation across venues (within 3 percentage points for most elements). NeurIPS shows slightly higher specification rates, possibly due to its larger multi-agent systems track. Given the small absolute gaps, no strong venue-level pattern is evident in this sample.

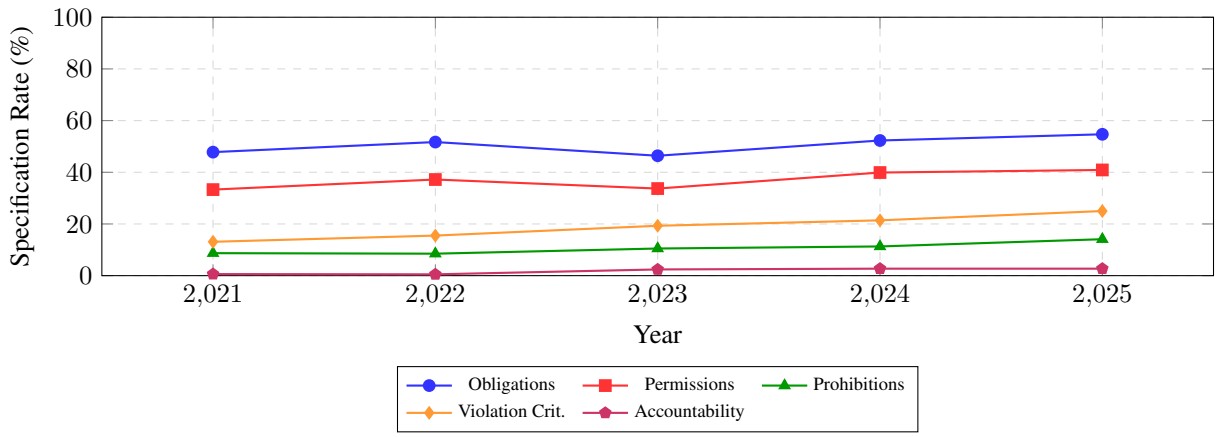

*Figure 3.* Agent Contract element specification rates by year (2021–2025).

## F.5. Agent Contract Element Analysis

In addition to Research Contract elements, we analyzed whether papers specify expected agent behaviors. Table 16 presents Agent Contract element specification rates by year.

*Table 16.* Agent Contract element specification rates by year (%).

| Year | N | Obligations | Permissions | Prohibitions | Violation Crit. | Accountability |
|------|------|-------------|-------------|--------------|-----------------|----------------|
| 2021 | 312 | 47.8 | 33.3 | 8.7 | 13.1 | 0.6 |
| 2022 | 387 | 51.7 | 37.2 | 8.5 | 15.5 | 0.5 |
| 2023 | 457 | 46.4 | 33.7 | 10.5 | 19.3 | 2.4 |
| 2024 | 514 | 52.3 | 39.9 | 11.3 | 21.4 | 2.7 |
| 2025 | 711 | 54.7 | 40.9 | 14.1 | 25.0 | 2.7 |
| **Change** | – | +6.9 | +7.6 | +5.4 | +11.9 | +2.1 |

*Table 17.* Agent Contract element specification rates overall (N=2,381).

| Element | Yes | No | Yes (%) |
|---------|-----|-----|---------|
| Obligations | 1,219 | 1,162 | 51.2 |
| Permissions | 898 | 1,483 | 37.7 |
| Prohibitions | 266 | 2,115 | 11.2 |
| Violation Criteria | 477 | 1,904 | 20.0 |
| Accountability Chain | 48 | 2,333 | 2.0 |

**Observations:** Obligations (51.2%) are most commonly specified, often implicitly through reward functions or task descriptions. Permissions (37.7%) are moderately specified, typically through action space definitions. Prohibitions (11.2%) are rarely explicit—most papers define what agents *can* do, not what they *must not* do. Violation Criteria (20.0%) shows the strongest improvement (+11.9pp), possibly due to increased interest in safe RL. Accountability Chain (2.0%) is nearly absent, explaining why real-world incidents result in vague attributions.

## F.6. Co-occurrence Analysis

We analyzed how often papers specify multiple elements together.

**Observations:** Joint specification of stakeholder and evaluation bounds appears in 18.7% of all papers. Relative to the 34.6% stakeholder rate, this implies $P(\text{Eval Bounds} \mid \text{Stakeholder}) \approx 54.0\%$, above the 24.4% marginal rate for Eval Bounds. Adversarial context specification correlates weakly with other elements. Auditability rarely co-occurs with any element, indicating it is systematically overlooked.

*Table 18.* Co-occurrence of Research Contract elements (percentage of papers specifying both).

|  | Stakeholder | Intervention | Eval Bounds | Adversarial | Auditability |
|---|---|---|---|---|---|
| Explanandum | 33.2 | 47.8 | 23.6 | 11.1 | 0.8 |
| Stakeholder | – | 24.1 | 18.7 | 7.3 | 0.6 |
| Intervention | – | – | 19.2 | 8.4 | 0.5 |
| Eval Bounds | – | – | – | 6.8 | 0.4 |
| Adversarial | – | – | – | – | 0.3 |

## F.7. Full Specification Analysis

We examined how many papers specify all six elements ("full specification") versus partial specifications.

*Table 19.* Distribution of specification completeness.

| Elements Specified | N Papers | Percentage |
|---|---|---|
| 0 elements | 142 | 6.0% |
| 1 element | 847 | 35.6% |
| 2 elements | 698 | 29.3% |
| 3 elements | 412 | 17.3% |
| 4 elements | 213 | 8.9% |
| 5 elements | 66 | 2.8% |
| 6 elements (full) | 3 | 0.1% |

**Key finding:** Only 3 papers (0.1%) in our corpus specify all six Research Contract elements. 70.9% of papers specify two or fewer elements. This quantifies the gap our framework addresses.

## F.8. Qualitative Examples

To illustrate what high and low specification looks like concretely, we provide two synthetic hypothetical examples. These examples are composite patterns based on the coding distributions in our corpus and are not taken from any single paper. The "Yes/No" assignments below follow the rubric in Table 12, and the examples are constructed to match the 5/6 and 1/6 specification profiles rather than cherry-picked individual papers.

F.8.1. HYPOTHETICAL WELL-SPECIFIED EXAMPLE (5 OF 6 ELEMENTS — TOP 3%)

- **Explanandum (Yes)**: "Why agent coalitions form in competitive resource allocation" — system-level, explicitly scoped.

- **Stakeholder (Yes)**: "Game designers seeking to balance multiplayer games" — specific audience with stated expertise context.

- **Intervention Unit (Yes)**: "Counterfactuals at the policy parameter level" — explicit granularity.

- **Evaluation Bounds (Yes)**: "$\geq 85\%$ fidelity to true coalition dynamics, validated on held-out episodes" — quantitative threshold with stated validity scope.

- **Adversarial Context (Yes)**: "Cooperative information sharing assumed; adversarial agents out of scope" — explicit threat exclusion.

- **Auditability (No)**: Not specified.

This level of specification enables direct comparison with other coalition-explanation methods, replication under stated bounds, and clear identification of what the method does *not* cover.

F.8.2. HYPOTHETICAL POORLY-SPECIFIED EXAMPLE (1 OF 6 ELEMENTS — REPRESENTATIVE OF 35.6%)

- **Explanandum (Yes)**: "Why negotiation fails during multi-robot resource contention" — explicit system phenomenon.

- **Stakeholder (No)**: Not specified.

- **Intervention Unit (No)**: Not specified.

- **Evaluation Bounds (No)**: "Our method outperforms baselines" — relative only, no threshold or validity scope.

- **Adversarial Context (No)**: Not specified.

- **Auditability (No)**: Not specified.

Without these specifications, a reader cannot determine who the explanation is for, what counterfactual questions it answers, under what conditions its claims hold, or how to compare it against the well-specified paper above. The contrast illustrates why specification gaps are not merely bureaucratic omissions but barriers to scientific comparison and deployment readiness.

### F.9. Limitations of Analysis

This analysis has several limitations. First, we analyzed only NeurIPS, ICML, and ICLR; papers from AAMAS, IJCAI, or domain-specific venues may show different patterns. Second, some coding decisions (especially "implicit" vs. "explicit" specification) involve judgment, and we did not compute inter-rater reliability. Third, the 2021–2025 temporal scope may not capture earlier foundational work or very recent developments. Fourth, papers accepted at top venues may have higher specification rates than rejected papers or preprints.

Despite these limitations, the analysis provides robust evidence that specification gaps are systematic and widespread, justifying the need for explicit contracts.

