# OpenReview forum: "Position: Multi-Agent Explainability Needs Contracts Before Methods"
_ICML.cc/2026/Position_Paper_Track — ICML 2026 Position Paper Track regular_

### Official Review · Reviewer_qR8i · 2026-03-13

**Significance:** 3
**Argument Clarity:** 3
**Rating:** 5
**Confidence:** 3

**Questions:**

Can you please show examples of "contracts" for a couple of existing methods to fully appreciate the value of contracts?

**Alternative Views Section:**

Yes

**Compliance With Llm Reviewing Policy A Conservative:**

Affirmed.

**Discussion Potential:**

3

**Final Justification:**

The authors have addressed my comments, including the request for a better formalization.

**Paper Summary:**

The paper presents the position that multi-agent explainability should document the expectations from an explanation before developing methods for the same. This is an interesting viewpoint.

**Position:**

Yes

**Position In Title:**

Yes

**Related Work:**

3

**Strengths And Weaknesses:**

Strengths:
1. There is no doubt that better documentation of both research and agent contracts, i.e., the purpose of the explanation, will assist the XAI and AI community.
2. The motivation using contemporary examples is interesting.
3. The analysis of published papers at top AI venues in this context is really reassuring.

Weaknesses:
1. The use of the term "contract" in this position paper is less mathematical that I initially anticipated. It would have been great to use a formal contract as a basis for reasoning.
2. The separation into research and agent contracts is not natural. Why only these two sets of contracts? How did we reach this design decision?
3. It would be good to see "contracts" for a couple of existing methods to fully appreciate the value of contracts.

**Support:**

3

---

> ### Author Rebuttal · Authors · 2026-03-28
>
> We thank the reviewer for their feedback. We are glad that the reviewer found the overall position interesting and the empirical motivation compelling. We address the points raised below:
>
> **Weakness 1:**
>
> **(1) On the request for stronger formalization:** We now formalize both contracts in the revised manuscript.
>
> **Section 3.1** now defines the Research Contract formally as:
>
> $$
> \mathcal{C}_R = (E, S, I, B, X, G)
> $$
>
> where $E$ is the explanandum, $S$ is stakeholder specification, $I$ is intervention unit, $B = \{(m_i, \theta_i, d_i)\}$ is evaluation bounds (metric, threshold, validity domain), $X$ is adversarial context, and $G$ is governance specification. This types the six Table 1 elements so they are checkable and comparable across papers.
>
> **Section 3.2** now defines the Agent Contract formally as:
>
> $$
> \mathcal{C}_A = (\mathcal{N}, \succ, \mathcal{V}, \mathcal{A})
> $$
>
> where:
>
> - **Norm set** ($\mathcal{N}$): A finite set of norms $n_i = (d_i, r_i, \phi_i, \alpha_i, \pi_i)$, $d_i \in \{O, P, F\}$, with bearer $r_i$, activation condition $\phi_i$, regulated action/state $\alpha_i$, and priority $\pi_i$.
> - **Priority ordering** ($\succ$): An ordering that resolves conflicts when active norms prescribe incompatible actions (Vasconcelos et al., 2009).
> - **Violation predicates** ($\mathcal{V}$): A set of predicates $v_i : S_{\texttt{obs}} \to \{\texttt{violated}, \texttt{satisfied}, \texttt{unknown}\}$, where $\texttt{unknown}$ captures partial observability.
> - **Accountability rules** ($\mathcal{A}$): A set of rules $a_k = (S_k, r_k, t_k)$, $t_k \in \{\texttt{primary}, \texttt{secondary}\}$, mapping violation patterns $S_k$ to responsible party $r_k$ and responsibility type $t_k$.
>
> This grounds the Agent Contract in deontic logic (von Wright, 1951) and normative MAS conflict handling (Vasconcelos et al., 2009).
>
> **(2) On the connection between formalism and paper semantics:** **Section 3.3** now formally defines:
>
> $$
> \mathcal{B}_{\text{expected}} = \{ \, b \mid \forall n_i \in \mathcal{N},\; v_i(b) \not= \texttt{violated} \, \}
> $$
>
> and $\Delta$ as the violated norms detected by $\mathcal{V}$. A concrete warehouse norm instantiation:
>
> $n_3 = (F, R_*, \top, \texttt{enter-wo-grant}, 3)$
>
> $n_4 = (O, \texttt{coord-sys}, \phi_c = \texttt{concurrent-requests}, \alpha_c = \texttt{consistent-granting}, 3)$
>
> with $n_3, n_4 \succ n_1$, so safety and coordination norms dominate procedural ones. Full walkthrough in **Appendix B.3** of the revised paper.
>
> **Weakness 2:**
>
> **(3) On why exactly two contracts:** Section 3.3 (Contract Interaction) explains how the Research Contract specifies *what* is explained, *for whom*, and *with what guarantees*, while the Agent Contract defines the behavioral baseline against which deviations are identified. Figure 1 illustrates their complementary roles.
>
> In the revised paper, we added a dedicated **"Why exactly two contracts?"** paragraph in Section 3.2 that makes this orthogonality explicit:
>
> > The two-contract decomposition reflects an orthogonality between epistemological and behavioral concerns. The Research Contract encodes what constitutes a valid explanation (target, audience, quantitative criteria, governance), while the Agent Contract encodes what constitutes expected behavior (obligations, permissions, prohibitions, and accountability). These concerns vary independently: the same MAS may require different explanations for different stakeholders while preserving one behavioral baseline, and the same explanation requirements may apply across differently specified agent systems. The case studies demonstrate both patterns. A unified contract would conflate explanation requirements with behavior specification and reduce reuse; splitting into more than two baseline contracts (for example, separate stakeholder and evaluation contracts) does not add independent structure because those elements are already components of $\mathcal{C}_R$. Two contracts are therefore the minimal decomposition that preserves this orthogonality.
>
> **Weakness 3:**
>
> **(4) On examples for existing methods:** In the revised paper, we added Section 4.5 ("Retroactive Contract Analysis of Existing Methods") and full coding details in Appendix B. We retroactively coded Rodriguez et al. (2025) (TriQPAN) and Gyevnar et al. (2023) (CEMA) against both contract families using the same rubric as our corpus analysis (Specified / Partial / Not specified). Two findings emerged: (i) even a specification-aware method like TriQPAN leaves critical deployment elements underspecified (prohibitions, accountability, adversarial context are all Not specified or Partial), and (ii) a technically strong causal analysis method like CEMA identifies influential interaction patterns without defining expected behavior norms — obligations, violation criteria, and accountability are all Not specified, so the method remains descriptive rather than norm-contrastive. This mirrors our corpus-level findings.

---

> > ### Author Rebuttal · Reviewer_qR8i · 2026-04-02
> >
> > The rebuttal is satisfactory, My revised recommendation of accept adequately reflects the current state of the paper.

---

### Official Review · Reviewer_AajE · 2026-03-13

**Significance:** 4
**Argument Clarity:** 3
**Rating:** 5
**Confidence:** 4

**Questions:**

1. To address the implementation burden, could the authors provide or envision lightweight software tools (e.g., a linter or template generator) to help researchers easily create these contracts and lower the adoption barrier?
2. How does the framework handle dynamic environments where the "Agent Contract" (obligations/permissions) might need to evolve or be renegotiated at runtime, rather than being statically defined pre-deployment?

**Alternative Views Section:**

Yes

**Compliance With Llm Reviewing Policy A Conservative:**

Affirmed.

**Discussion Potential:**

3

**Paper Summary:**

The paper argues that current research in Explainable AI for Multi-Agent Systems (MAS XAI) suffers from fragmentation and underspecification, leading to non-comparable and non-reproducible results disconnected from deployment needs. Based on a systematic analysis of 2,381 MAS-related papers from top machine learning venues (2021–2025), the authors identify critical gaps in practice, such as the widespread omission of stakeholder specifications (65%), quantitative evaluation bounds (76%), and auditability requirements (99%). To address this, the authors propose a framework requiring the explicit specification of two contracts before developing methods: a "Research Contract" (defining explanandum, stakeholder, intervention unit, evaluation bounds, adversarial context, and auditability) and an "Agent Contract" (defining expected behaviors via obligations, permissions, prohibitions, violation criteria, and accountability chains). By adopting these method-agnostic contracts, the paper contends that MAS XAI research can transform vague post-hoc descriptions into systematic and verifiable explanations.

**Position:**

Yes

**Position In Title:**

Yes

**Related Work:**

4

**Strengths And Weaknesses:**

Strengths:
1. The paper sharply identifies a fundamental problem in MAS XAI research—fragmentation, lack of standards, and disjointed methodologies ("talking past each other"). This clear articulation of the field's chaotic state is highly appropriate and impactful for a position paper, effectively drawing attention to the need for shared norms.
2. The authors support their claims with a rigorous systematic analysis of 2,381 papers from top venues (2021–2025). This large-scale survey provides undeniable quantitative evidence for the identified gaps (e.g., lack of auditability and evaluation bounds), moving the argument beyond mere opinion.
3. Beyond criticism, the paper proposes a concrete "Two Contracts" framework (Research and Agent Contracts). This proposal is well-structured, method-agnostic, and theoretically grounded in normative MAS literature, offering a practical path forward that is applicable to diverse architectures including modern LLM-based agents.


Weakness:
1. Requiring researchers to formally specify detailed contracts (especially the Agent Contract with all its deontic elements like violation criteria) for every study imposes a significant overhead. Without automated tools or standardized templates, this added friction might hinder widespread adoption, as researchers may view it as bureaucratic rather than value-adding.
2. Pre-defining comprehensive "obligations" and "prohibitions" is extremely challenging in complex, open-ended, or adaptive systems where behavior is designed to emerge from interactions. In such cases, rigid a priori specifications might be unrealistic or overly restrictive, potentially failing to capture the very emergent properties that researchers aim to study.

**Support:**

4

---

> ### Author Rebuttal · Authors · 2026-03-28
>
> We thank the reviewer for their feedback. We are glad that the reviewer found the paper's motivation, large-scale evidence, and framework potential compelling. We address the points raised below:
>
> **Weakness 1:**
>
> **(1) On adoption overhead:** Contracts need not be lengthy. As shown in Appendix B.3 (Minimal Specification Example) of the original paper, a submission-ready contract can be written in a few lines:
>
> ```
> MINIMAL RESEARCH CONTRACT
> ==========
> Explanandum: System-level coordination
>     failures in warehouse multi-robot systems
> Stakeholder: Operations managers conducting
>     post-incident analysis
> Intervention Unit: Agent actions (what if
>     robot R took action A' instead of A?)
> Evaluation: Causal accuracy >= 90% validated
>     against ground-truth logs
> Adversarial: Non-adversarial agents;
>     sensor noise <= 5%
> Auditability: Full action traces required;
>     90-day retention
> ```
>
> This five-line specification, while brief, provides sufficient information for reproduction and comparison. Full machine-readable YAML templates appear in Appendix B (Research Contract) and Appendix C (Agent Contract), with paper-submission checklists in Appendix B.2.
>
> We envision adoption occurring in phases: **lightweight adoption** (free-text contracts like the example above, minimal overhead) $\to$ **widespread adoption** (leading to community convergence on common elements) $\to$ **standardized adoption** (machine-readable templates based on these common elements, automated validators). The lightweight template above represents the first phase — minimal effort from researchers while still making specifications explicit and comparable. As this practice becomes more widely adopted, the needs of the MAS XAI field can be distilled into standardized contracts amenable to automation. No technical hurdles prevent this progression; documentation standards like Model Cards followed the same trajectory from academic proposal to platform integration. We have also expanded the Call to Action (Section 6) in the revised paper to go further in-depth into our envisioned adoption plan for our contracts.
>
> **(2) On tooling support:** While we do not yet include a full linter implementation, the revised YAML templates (Appendices B and C) are structured to be machine-readable and directly usable as a basis for validators and checkers, lowering the adoption burden in line with the lightweight adoption phase described above.
>
> **Weakness 2:**
>
> **(3) On dynamic/adaptive environments:** We acknowledge that the primary focus of our framework is not adaptive environments. However, contracts can accommodate dynamic settings through tiered mutability. In the revised paper, we expanded Alternative Views (Objection 3) with the following structure:
>
> - **Immutable safety invariants** — prohibitions on physical harm or unauthorized destructive actions remain fixed and occupy the highest-priority norms (e.g., Constitutional AI's "hardcoded" safety rules; Bai et al., 2022).
> - **Revisable coordination norms** — rules like intersection yielding or task reallocation protocols can be updated through controlled processes.
> - **Adaptive operational guidelines** — preferences like route selection or response formatting can adapt at runtime.
>
> In the formal Agent Contract $\mathcal{C}_A = (\mathcal{N}, \succ, \mathcal{V}, \mathcal{A})$ (defined in revised Section 3.2), each norm $n_i = (d_i, r_i, \phi_i, \alpha_i, \pi_i)$ carries a priority $\pi_i$. These tiers map directly to priority levels: safety invariants receive the highest $\pi_i$, coordination norms receive intermediate values, and operational guidelines receive the lowest. The ordering $\succ$ resolves conflicts so that lower-tier runtime adaptations cannot override higher-priority safety constraints. This aligns with existing practice: runtime norm revision mechanisms such as IRON (Vasconcelos et al., 2013) already support adding, removing, and revising norms during execution for lower-tier coordination updates.

---

> > ### Author Rebuttal · Reviewer_AajE · 2026-04-02
> >
> > The rebuttal clearly addressed my concerns about the complexities of practical operation and risk control.

---

### Official Review · Reviewer_pQdc · 2026-03-22

**Significance:** 2
**Argument Clarity:** 2
**Rating:** 4
**Confidence:** 3

**Questions:**

See weaknesses!

**Alternative Views Section:**

Yes

**Compliance With Llm Reviewing Policy A Conservative:**

Affirmed.

**Discussion Potential:**

2

**Paper Summary:**

This paper argues that current multi-agent explainability research focuses too much on methods while neglecting the need for clear specification of what explanations should achieve. It proposes a contract-based framework consisting of Research Contracts and Agent Contracts. Through literature analysis, quantitative studies, and case studies, the paper shows that the lack of such specifications makes existing explanations non-verifiable.

**Position:**

Yes

**Position In Title:**

Yes

**Related Work:**

2

**Strengths And Weaknesses:**

The paper correctly identifies a key gap that explainability should be specification-driven, not method-driven.

The contract abstraction is clear and actionable.

Although simple, the emperical analysis shows most works lack key specification element and accountability is largely missing.

======================================================

The contracts are not formally defined, not verifiable, and even not theoretically grounded.

There is no real system-level experimental validation (only case studies).

**Support:**

3

---

> ### Author Rebuttal · Authors · 2026-03-28
>
> We thank the reviewer for their feedback. We are glad that the reviewer found the core problem framing and contract abstraction clear and actionable. We address the points raised below:
>
> Weakness 1:
>
> (1) On the concern that contracts were not formally defined or theoretically grounded: we now formalize both contracts in the revised manuscript.
>
> **Section 3.1** now defines the Research Contract formally as:
>
> $$
> \mathcal{C}_R = (E, S, I, B, X, G)
> $$
>
> where $E$ is the explanandum, $S$ is stakeholder specification, $I$ is intervention unit, $B = \{(m_i, \theta_i, d_i)\}$ is evaluation bounds (metric, threshold, validity domain), $X$ is adversarial context, and $G$ is governance specification. This formalization types the six elements already used in Table 1 so they are checkable and comparable across papers without imposing heavy formal overhead.
>
> **Section 3.2** now defines the Agent Contract formally as:
>
> $$
> \mathcal{C}_A = (\mathcal{N}, \succ, \mathcal{V}, \mathcal{A})
> $$
>
> where:
>
> - **Norm set** ($\mathcal{N}$): A finite set of norms $n_i = (d_i, r_i, \phi_i, \alpha_i, \pi_i)$, $d_i \in \{O, P, F\}$, with bearer $r_i$, activation condition $\phi_i$, regulated action/state $\alpha_i$, and priority $\pi_i$.
> - **Priority ordering** ($\succ$): An ordering over norms that resolves conflicts when simultaneously active norms prescribe incompatible actions (Vasconcelos et al., 2009).
> - **Violation predicates** ($\mathcal{V}$): A set of predicates $v_i : S_{\texttt{obs}} \to \{\texttt{violated}, \texttt{satisfied}, \texttt{unknown}\}$, where $\texttt{unknown}$ captures partial observability.
> - **Accountability rules** ($\mathcal{A}$): A set of rules $a_k = (S_k, r_k, t_k)$, $t_k \in \{\texttt{primary}, \texttt{secondary}\}$, mapping violation patterns $S_k$ to responsible party $r_k$ and responsibility type $t_k$.
>
> This tuple grounds the five Agent Contract elements in a compact operational form based on deontic logic (von Wright, 1951) and normative MAS conflict handling (Vasconcelos et al., 2009).
>
> (2) On verifiability and operational linkage to existing notation: **Section 3.3** now formally defines:
>
> $$
> \mathcal{B}_{\text{expected}} = \{ \, b \mid \forall n_i \in \mathcal{N},\; v_i(b) \not= \texttt{violated} \, \}
> $$
>
> and $\Delta$ can be represented as the set of violated norms in $\mathcal{N}$ detected by $\mathcal{V}$. We also include a concrete norm tuple instantiation in the main text for the warehouse case, e.g.:
>
> $n_3 = (F, R_*, \top, \texttt{enter-wo-grant}, 3)$
>
> $n_4 = (O, \texttt{coord-sys}, \phi_c = \texttt{concurrent-requests}, \alpha_c = \texttt{consistent-granting}, 3)$
>
> with $n_3, n_4 \succ n_1$, so safety and coordination integrity dominate procedural norms. A full predicate-level walkthrough appears in **Appendix B.3**.
>
> Weakness 2:
>
> (3) On the concern about limited system-level validation: we agree that intervention-style deployment experiments would further strengthen the evidence for contracts. As a position paper, our contribution is the framework and the argument for why it is needed, rather than a full systems evaluation. That said, we have revised the paper and added a retroactive contract analysis of two existing MAS XAI methods (Section 4.5, with detailed coding in Appendix B), demonstrating that the contracts can be operationally applied to real methods. Specifically, we coded Rodriguez et al. (2025) (TriQPAN) and Gyevnar et al. (2023) (CEMA) against both contract families using a transparent rubric (Specified / Partial / Not specified). Two findings emerged: (i) even a specification-aware method like TriQPAN leaves critical deployment elements underspecified (prohibitions, accountability, adversarial context are all Not specified or Partial), and (ii) a technically strong causal analysis method like CEMA can identify influential interaction patterns without defining expected behavior norms, remaining descriptive rather than norm-contrastive (obligations, violation criteria, and accountability are all Not specified). These gaps mirror our corpus-level findings and are not visible from the original papers alone. While this is not a deployment experiment, it provides concrete evidence that the contracts are applicable beyond our own case studies and that they surface actionable gaps. Prospective system-level intervention experiments — deploying contracts alongside an active MAS and measuring whether they improve explanation quality, debugging efficiency, or accountability attribution — remain planned follow-up work.

---

> > ### Author Rebuttal · Reviewer_pQdc · 2026-04-05
> >
> > Thank the authors for the response!

---

### Decision · Program_Chairs · 2026-04-30

**Decision:**

Accept (regular)

**Comment:**

This is an exemplary position paper. It is thoughtful, well-supported, and forward-looking. The review process has sharpened its arguments and solidified its contribution.